# MOTIF-INDUCED GRAPH NORMALIZATION

## ABSTRACT

Graph Neural Networks (GNNs) have emerged as a powerful category of learning architecture for handling graph-structured data. However, the existing GNNs usually follow the neighborhood aggregation scheme, ignoring the structural characteristics in the node-induced subgraphs, which limits their expressiveness for the downstream tasks of both the graph- and node-level predictions. In this paper, we strive to strengthen the general discriminative capabilities of GNNs by devising a dedicated plug-and-play normalization scheme, termed as *Motif-induced Normalization* (MotifNorm), that explicitly considers the intra-connection information within each node-induced subgraph. To this end, we embed the motif-induced structural weights at the beginning and the end of the standard BatchNorm, as well as incorporate the graph instance-specific statistics for improved distinguishable capabilities. In the meantime, we provide the theoretical analysis to support that the MotifNorm scheme can help alleviate the over-smoothing issue, which is conducive to designing deeper GNNs. Experimental results on eight popular benchmarks across all the tasks of the graph, node, as well as link property predictions, demonstrate the effectiveness of the proposed method. Our code is made available in the supplementary material.

## 1 INTRODUCTION

In recent years, Graph Neural Networks (GNNs) have emerged as the mainstream deep learning architectures to analyze irregular samples where information is present in the form of graphs, which usually employs the message-passing aggregation mechanism to encode node features from local neighborhood representations (Kipf & Welling, 2017; Veličković et al., 2018; Xu et al., 2019; Yang et al., 2020b; Hao et al., 2021; Dwivedi et al., 2022b). As a powerful class of graph-relevant networks, these architectures have shown encouraging performance in various domains such as cell clustering (Li et al., 2022; Alghamdi et al., 2021), chemical prediction (Tavakoli et al., 2022; Zhong et al., 2022), social networks (Bouritsas et al., 2022; Dwivedi et al., 2022b), traffic networks (Bui et al., 2021; Li & Zhu, 2021), combinatorial optimization (Schuetz et al., 2022; Cappart et al., 2021), and power grids (Boyaci et al., 2021; Chen et al., 2022a).

However, the commonly used message-passing mechanism, i.e., aggregating the representations from neighborhoods, limits the expressive capability of GNNs to address the subtree-isomorphic phenomenon prevalent in the real world (Wijesinghe & Wang, 2022). As shown in Figure 1(a), subgraphs $S_{v_1}$, $S_{v_2}$ induced by $v_1$, $v_2$ are subtree-isomorphic, which decreases the GNNs' expressivity in graph-level and node-level prediction, i.e., (1) **Graph-level**: Straightforward neighborhood aggregations, ignoring the characterises of the node-induced subgraphs, lead to complete indistinguishability in the subtree-isomorphic case, which thus limits the GNNs' expressivity to be bottlenecked by the Weisfeiler-Leman (WL) (Weisfeiler & Leman, 1968) test. (2) **Node-level**: Under the background of over-smoothing (as illustrated in Figure 1(b)), the smoothing problem among the root representations of subtree-isomorphic substructures will become worser when aggregating the similar representations from their neighborhoods without structural characterises be considered.

In this paper, we strive to develop a general framework, compensating for the ignored characteristics among the node-induced structures, to improve the graph expressivity over the prevalent message-passing GNNs for various graph downstream tasks, e.g., graph, node and link predictions. Driven by the fact that deep models usually follow the CNA architecture, i.e., a stack of *convolution*, *normalization* and *activation*, where the normalization module generally follows GNNs convolution operations, we accordingly focus on developing a higher-expressivity generalized normalization

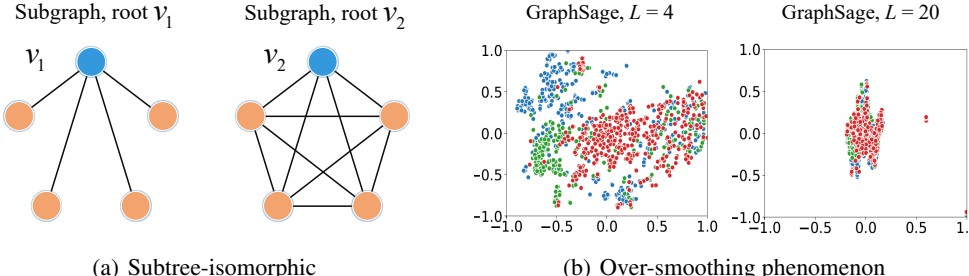

(a) Subtree-isomorphic        (b) Over-smoothing phenomenon

Figure 1: (a) The illustration of the subtree-isomorphic phenomenon, where two subgraphs $S_{v_1}$ and $S_{v_2}$ are induced by root node $v_1$ and $v_2$ with the same degree $k = 4$, but the connection information among neighborhoods is different. (b) The t-SNE illustration of over-smoothing issue on Cora dataset when GraphSage layer is up to 20. Here, we show the first three categories for visualization.

scheme to enhance the discriminative abilities of various GNNs' architectures. Thus, the question is, *"how to design such a powerful and general normalization module with the characteristics of node-induced substructures embedded ?"*

To address this challenge, this paper devises an innovative normalization mechanism, termed as *Motif-induced Normalization* (MotifNorm), that explicitly considers the intra-connection information in each node-induced subgraph (i.e., motif (Leskovec, 2021)) and embeds the achieved structural factors into the normalization module to improve the expressive power of GNNs. Specifically, we start by empirically disentangling the standard normalization into two stages, i.e., centering & scaling (CS) and affine transformation (AT) operations. We then concentrate on mining the intra-connection information in the node-induced subgraphs, and develop two elaborated strategies, termed as *representation calibration* and *representation enhancement*, to embed the achieved structural information into CS and AT operations. Eventually, we demonstrate that MotifNorm can generally improve the GNNs' expressivity for the task of graph, node and link predictions via extensive experimental analysis.

In sum, the contributions of this work can be summarized as follows:

- Driven by the conjecture that a higher-expressivity normalization with abundant graph structure power can generally strengthen the GNNs' performance, we develop a novel normalization scheme, termed as MotifNorm, to embed structural information into GNNs' aggregations.
- We develop two elaborated strategies, i.e., representation calibration and representation enhancement, tailored to embed the *motif-induced* structural factor into CS and AT operations for the establishment of MotifNorm in GNNs.
- We provide extensive experimental analysis on eight popular benchmarks across various domains, including graph, node and link property predictions, demonstrating that the proposed model is efficient and can consistently yield encouraging results.

It is worth mentioning that MotifNorm maintains the computational simplicity, which is beneficial to the model training for highly complicated tasks.

## 2    RELATED WORKS

In this section, we briefly introduce the existing normalization architectures in GNNs, which are commonly specific to the type of downstream tasks, i.e., graph-level and node-level tasks.

**Graph-level Normalization.** To address graph-level representation learning, Xu et al. (2019) adopt the standard BatchNorm (Ioffe & Szegedy, 2015) module to GIN as a plug-in component to stabilize the model's training. Based on the BatchNorm, Dwivedi et al. (2022a) normalize the node features with respect to the graph size to resize the feature space, and propose the ExpreNorm. To address the expressiveness degradation of GNNs for highly regular graphs, Cai et al. (2021) propose the GraphNorm with a learnable parameter for each feature dimension based on instance normalization. To adopt the advances of different normalizations, Chen et al. (2022c) propose UnityNorm by op-

timizing a weighted combination of four normalization techniques, which automatically selects a single best or the best combination for a specific task.

**Node-level Normalization.** This type of mechanism tries to rescale node representations for the purpose of alleviating over-smoothing issues (as shown in Figure 1(b)). Yang et al. (2020a) design the MeanNorm trick to improve GCN training by interpreting the effect of mean subtraction as approximating the Fiedler vector. Zhou et al. (2021) scale each node's features using its own standard deviation and proposes a variance-controlling technique, termed as NodeNorm, to address the variance inflammation caused by GNNs. To constrain the pairwise node distance, Zhao & Akoglu (2020b) introduce PairNorm to prevent node embeddings from over-smoothing on the node classification task. Furthermore, Liang et al. (2022) design ResNorm from the perspective of long-tailed degree distribution and add a shift operation in a low-cost manner to alleviate over-smoothing.

**However, these approaches are usually task-specific in GNNs' architectures, which means that they are not always significant in general contribution to various downstream tasks. Furthermore, the characteristics of node-induced substructures, which trouble GNNs' performance in various downstream tasks, are ignored by these normalizations.**

## 3    MOTIF-INDUCED GRAPH NORMALIZATION

In this section, we begin by give the preliminary with regard to our proposed MotfiNorm, along the way, introduce notations and a basis definition of motif-induced information.

Let $G = (V_G, E_G)$ denotes a undirected graph with $n$ vertices and $m$ edges, where $V_G = \{v_1, v_2, ..v_n\}$ is an set of vertices and $E_G$ is a unordered set of edges. $\mathcal{N}(v) = \{u \in V_G | (v, u) \in E_G\}$ denotes the neighbor set of vertex $v$, and its neighborhood subgraph $S_v$ is induced by $\tilde{\mathcal{N}}(v) = \mathcal{N}(v) \cup v$, which contains all edges in $E_G$ that have both endpoints in $\tilde{\mathcal{N}}(v)$. As shown in Figure 1(a), $S_{v_1}$ and $S_{v_2}$ are the induced subgraphs of $v_1$ and $v_2$. Their structural information are defined as:

**Definition 1.** *The motif-induced information $\xi(S_{v_i})$ denotes the structural weight of substructure $S_{v_i}$, i.e., node-induced subgraph (Leskovec, 2021) with regard to $v_i$, where $\xi$ is formulated as:*

$$\xi(S_{v_i}) = \varphi(|E_{S_{v_i}}|)\psi(|V_{S_{v_i}}|), \tag{1}$$

*where $| \cdot |$ denotes cardinality. $\varphi(|E_{S_{v_i}}|) = 2|E_{S_{v_i}}|/(|V_{S_{v_i}}| \cdot (|V_{S_{v_i}}| - 1))$ and $\psi(|V_{S_{v_i}}|) = |V_{S_{v_i}}|^2$ respectively refers to the density and power of $S_{v_i}$. This definition focuses more on edge information while two different subgraphs are subtree-isomorphic, on the contrary, focuses more on node power.*

### 3.1    PRELIMINARY

Batch normalization can be empirically divided into two stages, i.e., centering & scaling (CS) and affine transformation (AT) operations. For the input features $H \in \mathbb{R}^{n \times d}$, the CS and AT follow:

$$\begin{aligned} \text{CS} : H_{\text{CS}} &= \frac{H - \mathbb{E}(H)}{\sqrt{\mathbb{D}(H) + \epsilon}}, \\ \text{AT} : H_{\text{AT}} &= H_{\text{CS}} \odot \gamma + \beta, \end{aligned} \tag{2}$$

where $\odot$ is the dot product with the broadcast mechanism. $\mathbb{E}(H)$ and $\mathbb{D}(H)$ denote mean and variance statistics, and $\gamma, \beta \in \mathbb{R}^{1 \times d}$ are the learned scale and bias factors. In this work, we aim to embed structural weights into BatchNorm, and thus take a batch of graphs for example to give notations.

**Batch Graphs.** For a batch of graphs $\mathcal{G} = \{G_1, ..., G_m\}$ with node set $V_{\mathcal{G}} = V_{G_1} \cup V_{G_2}, ..., V_{G_m}$ $= \{v_1, ..., v_n\}$ and feature matrix $H \in \mathbb{R}^{n \times d}$. Motif-induced weights of this batch nodes is represented as $M_{\mathcal{G}} = \xi(S_{V_{\mathcal{G}}}) = [\xi(S_{V_{G_1}}); \xi(S_{V_{G_2}}); ...; \xi(S_{V_{G_m}})] = [\xi(S_{v_1}), ..., \xi(S_{v_n})] \in \mathbb{R}^{n \times 1}$. The segment summation-normalization $M_{\text{SN}} = [\mathcal{F}(\xi(S_{V_{G_1}})); \mathcal{F}(\xi(S_{V_{G_2}})); ...; \mathcal{F}(\xi(S_{V_{G_m}}))] \in \mathbb{R}^{n \times 1}$ where $\mathcal{F}(\xi(S_{V_{G_i}})) = \xi(S_{V_{G_i}})/\sum \xi(S_{V_{G_i}}) \in \mathbb{R}^{|V_{G_i}| \times 1}$, denotes the summation-normalization in each graph.

### 3.2    MOTIFNORM FOR GNNS

The commonly used message-passing aggregations are node-specific, i.e., ignoring the characteristics in the node-induced subgraphs, which limits the expressive capability of GNNs in various

downstream tasks. Therefore, we present a generalized graph normalization framework, termed as MotifNorm, to compensate for the ignored structural characteristics by GNNs, and develop two elaborated strategies: representation calibration (RC) and representation enhancement (RE).

**Representation Calibration (RC).** Before the CS stage, we calibrate the inputs by injecting the motif-induced weights as well as incorporate the graph instance-specific statistics into representations, which balances the distribution differences along with embedding structural information. For the input feature $H \in \mathbb{R}^{n \times d}$, the RC is formulated as:

$$\mathrm{RC} : \mathrm{H}_{\mathrm{RC}} = \mathrm{H} + w_{\mathrm{RC}} \odot \mathrm{H}_{\mathrm{SA}} \cdot (\mathrm{M}_{\mathrm{RC}} \mathbb{1}_d^T), \tag{3}$$

where $\mathbb{1}_d$ is the $d$-dimensional all-one cloumn vectors and $\cdot$ denotes the dot product operation. $w_{\mathrm{RC}} \in \mathbb{R}^{1 \times d}$ is a learned weight parameter. $\mathrm{M}_{\mathrm{RC}} = \mathrm{M}_{\mathrm{SN}} \cdot \mathrm{M}_{\mathcal{G}} \in \mathbb{R}^{n \times 1}$ is the calibration factor for RC, which is explained in details in Appendix A.3. $\mathrm{H}_{\mathrm{SA}} \in \mathbb{R}^{n \times d}$ is the segment averaging of H, obtained by sharing the average node features of each graph with its nodes, where each individual graph is called a segment in the DGL implementation (Wang et al., 2019).

**Representation Enhancement (RE).** Right after the CS operation, node features $\mathrm{H}_{\mathrm{CS}}$ are constrained into a fixed variance range and distinctive information is slightly weakened. Thus, we design the RE operation to embed motif-induced structural information into AT stage for the enhancement the final representations. The formulation of RE is written as follows:

$$\mathrm{RE} : \mathrm{H}_{\mathrm{RE}} = \mathrm{H}_{\mathrm{CS}} \cdot \mathrm{Pow}(\mathrm{M}_{\mathrm{RE}}, w_{\mathrm{RE}}), \tag{4}$$

where $w_{\mathrm{RE}} \in \mathbb{R}^{1 \times d}$ is a learned weight parameter, and $\mathrm{Pow}(\cdot)$ is the exponential function. To imitate affine weights in AT for each channel, we perform the segment summation-normlization on calibration factor $\mathrm{M}_{\mathrm{RC}}$ and repeats $d$ columns to obtain enhancement factor $\mathrm{M}_{\mathrm{RE}} \in \mathbb{R}^{n \times d}$, which ensures column signatures of $\mathrm{Pow}(\mathrm{M}_{\mathrm{RE}}, w_{\mathrm{RE}}) - 1$ are consistent.

**Expressivity Analysis.** MotfiNorm with injected RC and RE operation, compensating structural characteristics of subgraphs for GNNs, can generally improve graph expressivities as follows:

(1) Graph-level: For graph prediction tasks, MotfiNorm compensates for the structural information to distinguish the subtree-isomorphic case that 1-WL can not recognize, which could extend the GNNs more expressive than the 1-WL test. Specifically, an arbitrary GNN equipped with MotfiNorm is capable of more expressive abilities than the 1-WL test in distinguishing $k$-regular graphs.

(2) Node-level: The ignored structural information strengthens the node representations, which is beneficial to the downstream recognition tasks. Furthermore, MotifNorm with structural weights injected can help alleviate the over-smoothing issue, which is analyzed in the following Theorem 1.

(3) Training stability: The RC operation is beneficial to stabilize model training, which makes normalization operation less reliant on the running means and balances the distribution differences by considering the graph instance-specific statistics and is analyzed in the following Proposition 1.

**Theorem 1.** *MotifNorm helps alleviate the oversmoothing issue.*

*Proof.* Given two extremely similar embeddings of $u$ and $v$ (i.e., $\|\mathrm{H}_u - \mathrm{H}_v\|_2 \leq \epsilon$). Assume for simplicity that $\|\mathrm{H}_u\|_2 = \|\mathrm{H}_v\|_2 = 1$, $\|w_{\mathrm{RC}}\|_2 \geq c$, and the motif-induced information scores between $u$ and $v$ differs a considerable margin $\|(\mathrm{M}_{\mathrm{RC}} \mathbb{1}_d^T)_u - (\mathrm{M}_{\mathrm{RC}} \mathbb{1}_d^T)_v\|_2 \geq 2\epsilon/c$. We can obtain

$$\|(\mathrm{H}_u + (w_{\mathrm{RC}} \odot (\mathrm{M}_{\mathrm{RC}} \mathbb{1}_d^T))_u \cdot \frac{\mathrm{H}_u + \mathrm{H}_v}{2}) - (\mathrm{H}_v + (w_{\mathrm{RC}} \odot (\mathrm{M}_{\mathrm{RC}} \mathbb{1}_d^T))_v \cdot \frac{\mathrm{H}_v + \mathrm{H}_u}{2})\|_2$$

$$\geq -\|\mathrm{H}_u - \mathrm{H}_v\|_2 + \|((w_{\mathrm{RC}} \odot (\mathrm{M}_{\mathrm{RC}} \mathbb{1}_d^T))_u - (w_{\mathrm{RC}} \odot (\mathrm{M}_{\mathrm{RC}} \mathbb{1}_d^T))_v) \cdot \frac{\mathrm{H}_u + \mathrm{H}_v}{2}\|_2$$

$$\geq -\epsilon + \|w_{\mathrm{RC}}\|_2 \cdot \|(\mathrm{M}_{\mathrm{RC}} \mathbb{1}_d^T)_u - (\mathrm{M}_{\mathrm{RC}} \mathbb{1}_d^T)_v\|_2 \cdot \|\frac{\mathrm{H}_u + \mathrm{H}_v}{2}\|_2$$

$$\geq -\epsilon + 2\epsilon = \epsilon,$$

where the subscripts $u$, $v$ denote the $u$-th and $v$-th row of matrix $\in \mathbb{R}^{n \times d}$. This inequality demonstrates that our RC operation could differentiate two nodes by a margin $\epsilon$ even when their node embeddings become extremely similar after $L$-layer GNNs. Similarly, RE operation can also differentiates the embeddings, and the theoretical analysis is provided in A.1 □

**Proposition 1.** *RC operation is beneficial to stabilzing the model training.*

*Proof.* The complete proof is provided in Appendix A.2. □

### 3.3 THE IMPLEMENTATION OF MOTIFNORM

We merge RE operation into AT for a simpler formultation. Given the input feature $H \in \mathbb{R}^{n \times d}$, the formulation of the MotifNorm is written as:

$$
\begin{aligned}
\text{RC} &: H_{\text{RC}} = H + w_{\text{RC}} \odot H_{\text{SA}} \cdot (M_{\text{RC}} \mathbb{1}_d^T), \\
\text{CS} &: H_{\text{CS}} = \frac{H_{\text{RC}} - \mathbb{E}(H_{\text{RC}})}{\sqrt{\mathbb{D}(H_{\text{RC}}) + \epsilon}}, \\
\text{AT} &: H_{\text{AT}} = H_{\text{CS}} \cdot (\gamma + \mathbb{P})/2 + \beta,
\end{aligned}
\tag{5}
$$

where $\mathbb{P} = \text{Pow}(M_{\text{RE}}, w_{\text{RE}})$ and $H_{\text{AT}}$ is the output of MotifNorm. To this end, we add RC and RE operations at the beginning and ending of the original BatchNorm layer to strengthen the expressivity power after GNNs' convolution. The additional RC and RE operations are the dot product in $\mathbb{R}^{n \times d}$, thus their time complexity is $\mathcal{O}(nd)$.

## 4 EXPERIMENTS

To demonstrate the effectiveness of the proposed MotifNorm in different GNNs, we conduct experiments on three types of graph tasks, including graph-, node- and link-level predictions.

**Benchmark Datasets.** Eight datasets are employed in three types of tasks, including (i) Graph predictions: IMDB-BINARY, ogbg-moltoxcast, ogbg-molhiv, and ZINC. (ii) Node predictions: Cora, Pubmed and ogbn-proteins. (iii) Link predictions: ogbl-collab. Their basic statistics are summarized in Table 1.

Table 1: The statistic information of the benchmark datasets under different graph-structured tasks.

| Dataset name | Task level | #Graphs | Avg. #nodes | Avg. #edges |
|---|---|---|---|---|
| IMDB-BINARY | graph | 1,000 | 19.8 | 193.1 |
| ogbg-moltoxcast | graph | 8,576 | 18.8 | 19.3 |
| ogbg-molhiv | graph | 41,127 | 25.5 | 27.5 |
| ZINC | graph | 10,000 | 23.2 | 49.8 |
| Cora | node | 1 | 2,708 | 5,429 |
| Pubmed | node | 1 | 19,717 | 44,338 |
| ogbn-proteins | node | 1 | 132,534 | 39,561,252 |
| ogbl-collab | link | 1 | 235,868 | 1,285,465 |

**Baseline Methods.** We compare our MotifNorm to various types of normalization baselines for GNNs, including Batch-Norm (Ioffe & Szegedy, 2015), Uni-tyNorm (Chen et al., 2022c), Graph-Norm (Cai et al., 2021), ExpreNorm (Dwivedi et al., 2022a) for graph predictions, and Group-Norm (Zhou et al., 2020), PairNorm (Zhao & Akoglu, 2020a), MeanNorm (Yang et al., 2020a), NodeNorm (Zhou et al., 2021) for all three tasks.

More details about the datasets, baselines and experimental setups are provided in Appendix B.1.

In the following experiments, we aim to answer the questions: (i) Can MotifNorm improve the expressivity for graph isomorphism test, especially go beyond 1-WL on $k$-regular graphs? (Section 4.1) (ii) Can MotifNorm help alleviate the over-smoothing issue? (Section 4.2) (iii) Can MotifNorm generalize to various graph tasks? (Section 4.3)

### 4.1 EXPERIMENTAL ANALYSIS ON GRAPH ISOMORPHISM TEST

The IMDB-BINARY is a well-known graph isomorphism test dataset consisting of various $k$-regular graphs, which has become a common-used benchmark for evaluating the expressivity of GNNs. To make the training, valid and test sets follow the same distribution as possible, we adopt a hierarchical dataset splitting strategy based on the structural statistics of graphs (More detailed description are provided in Appendix B.1). For graph isomorphism test, Graph Isomorphism Network (GIN) (Xu et al., 2019) is known to be as powerful as 1-WL. Notably, GIN consists a neighborhood aggregation operation and a multi-layer perception layer (MLP), and this motivates a comparison experiment: comparing a one-layer MLP+MotifNorm with one-layer GIN to directly demonstrate MotifNorm's expressivities in distinguishing $k$-regular graphs.

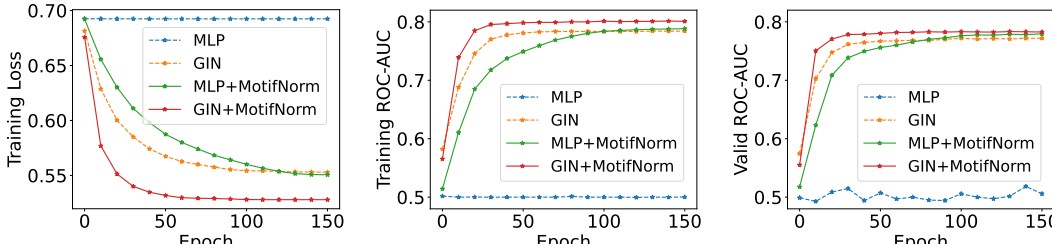

Figure 2: Learning curves of one-layer MLP, GIN, MLP + MotifNorm and GIN + MotifNorm on IMDB-BINARY dataset with various $k$-regular graphs.

Table 2: Experimental results on IMDB-BINARY dataset with various $k$-regular graphs. The best results under different backbones are highlighted with **boldface.**

| | Normalization | | Layers | ROC-AUC | | LOSS | |
| | Batchnorm | MotifNorm | | Test Split | Valid Split | Test Split | Valid Split |
|---|---|---|---|---|---|---|---|
| MLP | — | — | 1 | $56.66 \pm 1.09$ | $57.05 \pm 0.81$ | 0.6961 | 0.6961 |
| | ✓ | — | 1 | $56.86 \pm 0.72$ | $57.83 \pm 0.84$ | 0.6954 | 0.6954 |
| | — | ✓ | 1 | $\mathbf{78.16 \pm 0.52}$ | $\mathbf{78.69 \pm 0.44}$ | 0.5631 | 0.5581 |
| GIN | — | — | 1 | $77.40 \pm 0.20$ | $77.69 \pm 0.13$ | 0.5684 | 0.5658 |
| | ✓ | — | 1 | $77.71 \pm 0.19$ | $78.11 \pm 0.10$ | 0.5655 | 0.5593 |
| | — | ✓ | 1 | $\mathbf{78.42 \pm 0.15}$ | $\mathbf{78.89 \pm 0.15}$ | 0.5584 | 0.5565 |
| GSN | ✓ | — | 1 | $77.50 \pm 0.18$ | $77.54 \pm 0.16$ | 0.5696 | 0.5688 |
| | — | ✓ | 1 | $\mathbf{78.18 \pm 0.23}$ | $\mathbf{78.13 \pm 0.11}$ | 0.5637 | 0.5633 |
| GraphSNN | ✓ | — | 1 | $77.52 \pm 0.16$ | $77.61 \pm 0.18$ | 0.5691 | 0.5671 |
| | — | ✓ | 1 | $\mathbf{78.24 \pm 0.30}$ | $\mathbf{78.06 \pm 0.15}$ | 0.5631 | 0.5636 |
| GIN | ✓ | — | 4 | $78.41 \pm 0.21$ | $78.76 \pm 0.17$ | 0.5625 | 0.5590 |
| | — | ✓ | 4 | $\mathbf{79.55 \pm 0.35}$ | $\mathbf{79.62 \pm 0.38}$ | 0.5539 | 0.5510 |
| GCN | ✓ | — | 4 | $76.75 \pm 1.31$ | $76.96 \pm 0.69$ | 0.5755 | 0.5640 |
| | — | ✓ | 4 | $\mathbf{78.78 \pm 1.01}$ | $\mathbf{79.03 \pm 0.61}$ | 0.5597 | 0.5504 |
| GAT | ✓ | — | 4 | $75.10 \pm 1.51$ | $75.95 \pm 1.27$ | 0.5866 | 0.5852 |
| | — | ✓ | 4 | $\mathbf{78.87 \pm 0.80}$ | $\mathbf{79.20 \pm 0.56}$ | 0.5559 | 0.5448 |
| GSN | ✓ | — | 4 | $78.90 \pm 0.63$ | $79.28 \pm 0.70$ | 0.5555 | 0.5543 |
| | — | ✓ | 4 | $\mathbf{79.22 \pm 0.71}$ | $\mathbf{79.70 \pm 0.59}$ | 0.5536 | 0.5527 |
| GraphSNN | ✓ | — | 4 | $79.16 \pm 0.67$ | $79.35 \pm 0.82$ | 0.5541 | 0.5530 |
| | — | ✓ | 4 | $\mathbf{79.89 \pm 0.74}$ | $\mathbf{79.93 \pm 0.77}$ | 0.5517 | 0.5506 |

As illustrated in Figure 2, the vanilla MLP cannot capture any structural information and perform poorly, while the proposed MotifNorm method successfully improve the performance of MLP and even exceeds the vanilla GIN. Furthermore, Table 2 provides the quantitative comparison results, where GSN (Bouritsas et al., 2022) and GraphSNN (Wijesinghe & Wang, 2022) are two recent popular methods realizing the higher expressivity than the 1-WL. From these comparison results, the performance of one-layer MLP with MotifNorm is better than that of one-layer GIN, GSN, and GraphSNN. Moreover, the commonly used GNNs equipped with MotifNorm, e.g., GCN and GAT, achieve higher ROC-AUC than GIN when the layer is set as 4. GIN with MotifNorm achieves better performance and even goes beyond the GSN and GraphSNN. Furthermore, MotifNorm can further enhance the expressivity of GSN and GraphSNN. More detailed results on IMDB-BINARY are provided in Appendix B.2.

## 4.2 EXPERIMENTAL ANALYSIS ON THE OVER-SMOOTHING ISSUE

To show the effectiveness of MotifNorm for alleviating the over-smoothing issue in GNNs, we visualize the first three categories of Cora dataset in 2D space for a better illustration. We select PairNorm, NodeNorm, MeanNorm, GroupNorm and BatchNorm for comparison and set the number of layer as 32. Figure 3(a)~3(f) show the t-SNE visualization of different normalization techniques, and we can find that none of them suffer from the over-smoothing issue like Graph-Sage with BatchNorm (shown in Figure 1(b)). However, MotifNorm can better distinguish different

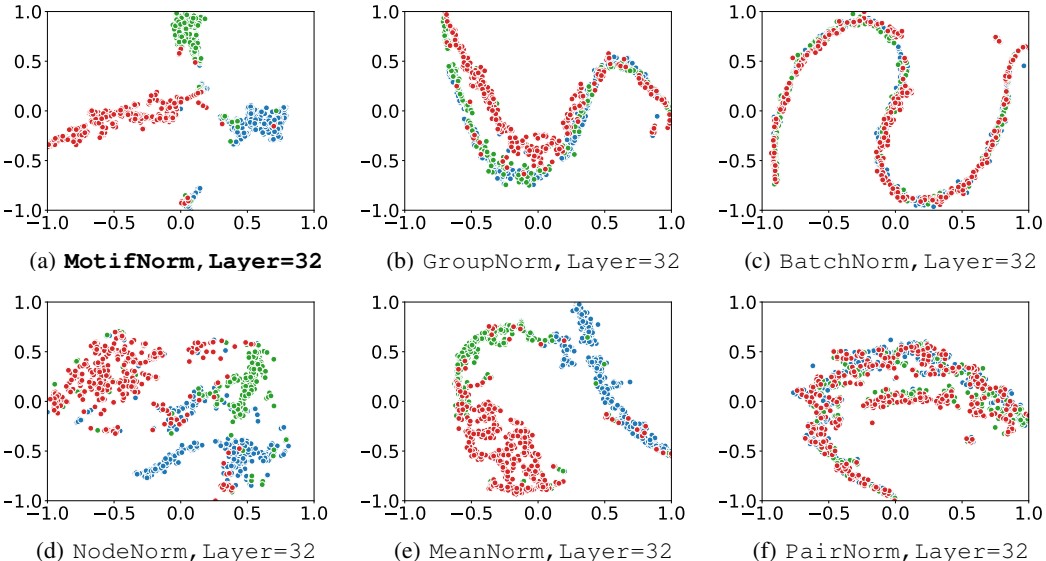

Figure 3: The t-SNE visualization of node representations using GCN with different normalization methods on Cora dataset.

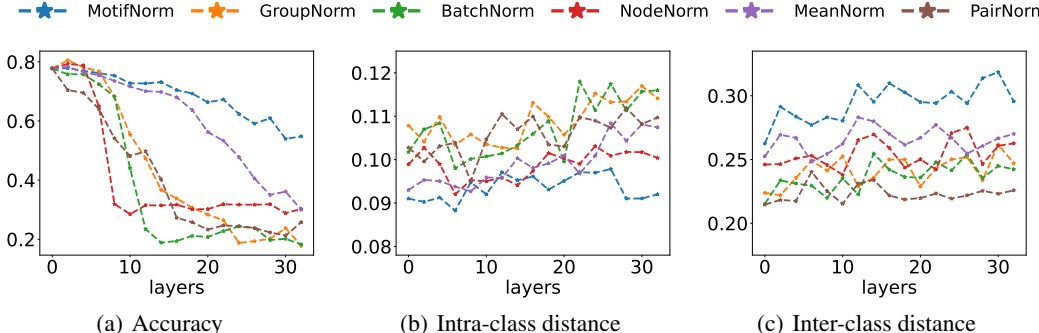

Figure 4: Experimental results of GCN with different normalization methods on Cora dataset.

categories into different clusters, i.e., the other normalization methods may lead to the loss of discriminant information and make the representations entangled.

Furthermore, we provide the quantitative results by considering three metrics including accuracy, intra-class distance and inter-class distance. In details, we set layers from 2 to 32 with the step size as 2, and visualize the line chart in Figure 4(a)~4(c). Figure 4(a) shows the accuracy with regard to layers, which directly demonstrates the superiority of MotifNorm when GNNs go deeper. In order to characterize the disentangling capability of different normalizations, we calculate the intra-class distance and inter-class distance with layers increasing in Figure 4(b) and 4(c). As shown in these two figures, MotifNorm obtains lower intra-class distance and higher inter-class distance, which means that the proposed MotifNorm enjoys better disentangling ability. We provide more t-SNE visualizations of different layer number and different GNNs' backbones in Appendix B.3.

## 4.3 More comparisons on the Other Six Datasets

For graph prediction task, we compare normalizations on ogbg-moltoxcast, ogbg-molhiv and ZINC by using GCN and GAT as backbones, where ZINC is a graph regression dataset. For node and link property predictions we conduct experiments on one social network dataset (Pubmed), one protein-protein association network dataset (ogbn-proteins) and a collaboration network between authors (ogbl-collab) by using GCN as backbone. The details are as follow:

Table 3: Experimental results of different normalization methods on the graph prediction task. We use GCN, GAT as the backbones. The best results on each dataset are highlighted with **boldface.**

| | Methods | ogbg-moltoxcast | | | ogbg-molhiv | | | ZINC | | |
| --- | --- | --- | --- | --- | --- | --- | --- | --- | --- | --- |
| | | $l=4$ | $l=16$ | $l=32$ | $l=4$ | $l=16$ | $l=32$ | $l=4$ | $l=16$ | $l=32$ |
| GCN | NoNorm | 61.13 | 59.34 | 56.08 | 76.01 | 71.90 | 60.59 | 0.643 | 0.690 | 0.748 |
| | GraphNorm | 60.78 | 53.75 | 53.36 | 75.59 | 65.55 | 66.49 | 0.592 | 0.655 | 1.547 |
| | BatchNorm | 63.39 | 59.73 | 53.47 | 76.11 | 76.62 | 74.21 | 0.573 | 0.611 | 0.655 |
| | UnityNorm | 63.86 | 61.94 | 59.18 | 75.94 | 72.14 | 69.44 | 0.552 | 0.576 | 0.650 |
| | ExpreNorm | 64.97 | 57.91 | 57.82 | 76.05 | 76.75 | 72.36 | 0.564 | 0.570 | 0.646 |
| | **MotifNorm** | **66.92** | **65.19** | **63.40** | **77.29** | **77.71** | **75.99** | **0.489** | **0.524** | **0.523** |
| GAT | NoNorm | 62.61 | 50.84 | 50.12 | 76.71 | 57.38 | 50.64 | 0.714 | 1.541 | 1.547 |
| | GraphNorm | 60.53 | 52.79 | 53.22 | 75.30 | 73.86 | 64.03 | 0.576 | 1.254 | 1.537 |
| | BatchNorm | 63.31 | 53.39 | 53.24 | 76.07 | 76.87 | 73.74 | 0.585 | 0.624 | 0.643 |
| | UnityNorm | 63.47 | 58.76 | 57.13 | 75.91 | 76.19 | 75.46 | 0.563 | 0.621 | 0.777 |
| | ExpreNorm | 65.56 | 57.65 | 57.60 | 76.99 | 72.24 | 72.56 | 0.555 | 0.562 | 1.451 |
| | **MotifNorm** | **66.57** | **64.04** | **58.26** | **77.36** | **77.08** | **76.70** | **0.495** | **0.517** | **0.522** |

Table 4: The comparison results of different normalization methods on the node prediction and link prediction task by using GCN as the backbone. The best results are highlighted with **boldface.**

| | Settings | Pubmed | | | ogbn-proteins | | | ogbl-collab | | |
| --- | --- | --- | --- | --- | --- | --- | --- | --- | --- | --- |
| | | $l=4$ | $l=16$ | $l=32$ | $l=4$ | $l=16$ | $l=32$ | $l=4$ | $l=16$ | $l=32$ |
| GCN | NoNorm | 76.16 | 54.67 | 45.58 | 69.16 | 63.24 | 63.15 | 35.38 | 22.11 | 15.24 |
| | PairNorm | 74.25 | 56.24 | 55.13 | 69.28 | 63.15 | 63.00 | 31.26 | 23.22 | 14.69 |
| | NodeNorm | 76.02 | 40.87 | 41.18 | 70.17 | 63.50 | 63.23 | 27.48 | 08.48 | 08.28 |
| | MeanNorm | 76.05 | 73.40 | 65.34 | 69.14 | 63.05 | 62.40 | 33.28 | 22.56 | 16.16 |
| | GroupNorm | 76.19 | 63.55 | 54.84 | 70.25 | 62.74 | 63.63 | 35.28 | 27.41 | 20.27 |
| | BatchNorm | 75.62 | 48.88 | 43.28 | 69.96 | 67.36 | 63.86 | 47.57 | 26.14 | 21.68 |
| | **MotifNorm** | **77.08** | **76.66** | **67.81** | **71.69** | **68.66** | **68.05** | **51.65** | **50.01** | **47.65** |

Table 5: Comparisons with empirical tricks on ogbg-molhiv and ZINC datasets.

| | Methods | ogbg-molhiv | ZINC |
| --- | --- | --- | --- |
| GCN | NoNorm | $77.21 \pm 0.430$ | $0.473 \pm 0.006$ |
| | UnityNorm | $77.56 \pm 1.060$ | $0.458 \pm 0.009$ |
| | ExpreNorm | $77.99 \pm 0.545$ | $0.436 \pm 0.008$ |
| | GraphNorm | $78.10 \pm 1.115$ | $0.396 \pm 0.008$ |
| | BatchNorm | $78.07 \pm 0.782$ | $0.398 \pm 0.003$ |
| | **MotifNorm** | $\mathbf{78.76 \pm 0.371}$ | $\mathbf{0.358 \pm 0.009}$ |

Table 6: Comparisons with empirical tricks on ogbn-proteins and ogbl-collab datasets.

| | Methods | ogbn-proteins | ogbl-collab |
| --- | --- | --- | --- |
| GCN | PairNorm | $69.84 \pm 0.533$ | $47.75 \pm 0.190$ |
| | NodeNorm | $72.53 \pm 1.514$ | $48.28 \pm 1.100$ |
| | MeanNorm | $71.09 \pm 1.236$ | $47.27 \pm 0.849$ |
| | GroupNorm | $73.17 \pm 0.503$ | $45.25 \pm 1.206$ |
| | BatchNorm | $72.39 \pm 0.611$ | $49.44 \pm 0.750$ |
| | **MotifNorm** | $\mathbf{73.55 \pm 1.271}$ | $\mathbf{52.15 \pm 0.647}$ |

Firstly, we adopt the vanilla GNN model without any tricks (e.g., residual connection, etc.), and provide the settings of the hyperparameter in Appendix B.1. Accordingly to the mean results (w.r.t., 10 different seeds) shown in Table 3 and Table 4, we can conclude that MotifNorm generally improves the graph expressivity of GNNs for graph prediction task and help alleviate the over-smoothing issue with the increase of layers. Furthermore, we provide more comparison experiments by using GIN and GraphSage as backbones in Appendix B.4.

Secondly, we perform empirical tricks in GNNs for a further comparison when generally obtaining better performances. The details of tricks on different datasets: (1) ogbg-molhiv: convolution with edge features, without input dropout, hidden dimension as 300, weight decay in {5e-5, 1e-5}, residual connection, GNN layers as 4. (2) ZINC: without input and hidden dropout, hidden dim as 145, residual connection, GNN layers as 4. (3) ogbn-proteins: without input and hidden dropout, hidden dim as 256, GNN layers as 2. (4) ogbl-collab: initial connection (Chen et al., 2020), GNN layers as 4. The results in Table 5 and Table 6 demonstrate that MotifNorm preserves the superiority in graph, node and link prediction tasks compared with other existent normalizations.

### 4.4 ABLATION STUDY AND DISCUSSION

To explain the superior performance of MotifNorm, we perform extensive ablation studies to evaluate the contributions of its two key components, i.e., representation calibration (RC) and represen-

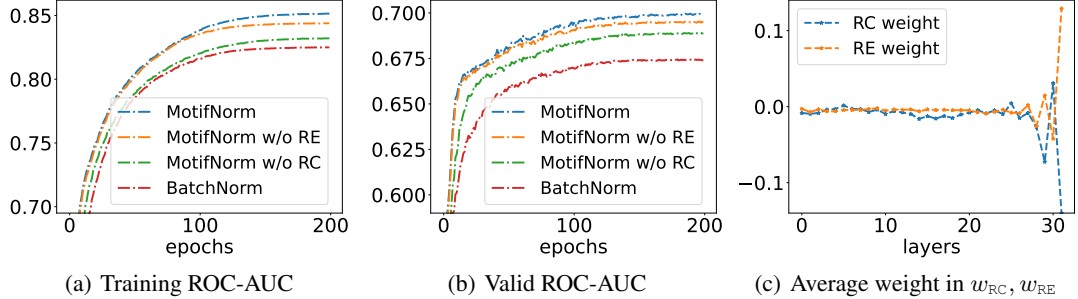

(a) Training ROC-AUC      (b) Valid ROC-AUC      (c) Average weight in $w_{\text{RC}}, w_{\text{RE}}$

Figure 5: Abalation study of the Representation Calibration (RC) and Representation Enhancement (RE) operations in MotifNorm on the ogbg-moltoxcast dataset. Here we use GCN as the basic backbone to conduct the abalation study.

tation enhancement (RE) operations. Firstly, we show in Figure 5 the ablation performance of GCN on ogbg-moltoxcast. Figure 5(a) and 5(b) show the ROC-AUC results with regard to training epochs when the layer number is set to 4, which show that both RC and RE can improve the classification performance. Furthermore, by comparing these two figures, RC performs better than RE in terms of recognition results, which plugs at the beginning of BatchNorm with the graph instance-specific statistics embedded.

To further explore the significance of RC and RE at different layers, we compute the mean values of $w_{\text{RC}}$ and $w_{\text{RE}}$, which are visualized in Figure 5(c). As can be seen from the mean statistis of $w_{\text{RC}}$, $w_{\text{RE}}$ among 32 layers' GCN, the absolute values of $w_{\text{RC}}$ and $w_{\text{RE}}$ become larger when the network goes deeper (especially in the last few layers), indicating that the structural information becomes more and more critical with the increase numbers of layers.

To evaluate the additional cost of RC and RE operations, we provide the runtime, parameter and memory comparison by using GCN ($l = 4$) with BatchNorm and MotfiNorm on the ogbg-molhiv dataset. Here, we provide the cost of runtime and memory by performing the code on NVIDIA A40. The cost information is provided in Table 7.

Table 7: The cost comparisons between BatchNorm and MotifNorm.

|  | runtime | parameter | memory |
|---|---|---|---|
| BatchNorm | 15.2s/epoch | 291.6K | 2305M |
| MotifNorm | 22.6s/epoch | 293.2K | 2347M |

**Discussion.** The main contribution of this work is to propose a more expressive normalization module, which can be plugged into any GNN architecture. Unlike existing normalization methods that are usually task-specific and also without substructure information, the proposed method explicitly considers the subgraph information to strengthen the graph expressivity across various graph tasks. In particular, for the task of graph classification, MotifNorm extends GNNs beyond the 1-WL test in distinguishing $k$-regular graphs. On the other hand, when the number of GNNs' layers becomes larger, MotifNorm can help alleviate the over-smoothing problem and meanwhile maintain better discriminative power for the node-relevant predictions.

## 5 CONCLUSION

In this paper, we introduce a higher-expressivity normalization architecture with an abundance of graph structure-specific information embedded to generally improve GNNs' expressivities and representatives for various graph tasks. We first empirically disentangle the standard normalization into two stages, i.e., centering & scaling (CS) and affine transformation (AT) operations, and then develop two skillful strategies to embed the subgraph structural information into CS and AT operations. Finally, we provide a theoretical analysis to support that MotifNorm can extend GNNs beyond the 1-WL test in distinguishing $k$-regular graphs and exemplify why it can help alleviate the over-smoothing issue when GNNs go deeper. Experimental results on 10 popular benchmarks show that our method is highly efficient and can generally improve the performance of GNNs for various graph tasks.

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

# A    THEOREM ANALYSIS

This section provides the corresponding proofs to support theorems in the main context.

## A.1    PROOF FOR THEOREM 1

**Theorem 1.** *MotifNorm helps alleviate the oversmoothing issue.*

*Proof.* Given two extremely similar embeddings of $u$ and $v$ (i.e., $\|H_u - H_v\|_2 \leq \epsilon$). Assume for simplicity that $\|H_u\|_2 = \|H_v\|_2 = 1$, $\|w_{RC}\|_2 \geq c_1$, and the motif-induced information scores between $u$ and $v$ differs a considerable margin $\|(M_{RC}\mathbb{1}_d^T)_u - (M_{RC}\mathbb{1}_d^T)_v\|_2 \geq 2\epsilon/c_1$. We can obtain

$$\|(H_u + (w_{RC} \odot (M_{RC}\mathbb{1}_d^T))_u \cdot \frac{H_u + H_v}{2}) - (H_v + (w_{RC} \odot (M_{RC}\mathbb{1}_d^T))_v \cdot \frac{H_v + H_u}{2})\|_2$$

$$\geq -\|H_u - H_v\|_2 + \|((w_{RC} \odot (M_{RC}\mathbb{1}_d^T))_u - (w_{RC} \odot (M_{RC}\mathbb{1}_d^T))_v) \cdot \frac{H_u + H_v}{2}\|_2$$

$$\geq -\epsilon + \|w_{RC}\|_2 \cdot \|(M_{RC}\mathbb{1}_d^T)_u - (M_{RC}\mathbb{1}_d^T)_v\|_2 \cdot \|\frac{H_u + H_v}{2}\|_2$$

$$\geq -\epsilon + 2\epsilon = \epsilon,$$

where the subscripts $u$, $v$ denote the $u$-th and $v$-th row of matrix $\in \mathbb{R}^{n \times d}$. This inequality demonstrates that our RC operation could differentiate two nodes by a margin $\epsilon$ even when their node embeddings become extremely similar after $L$-layer GNNs. Similarly, by assuming $\|Pow(M_{RE}, w_{RE})_u\|_2 \leq c_2$ and $\|Pow(M_{RE}, w_{RE})_u - Pow(M_{RE}, w_{RE})_v\|_2 \geq (1 + c_2) \cdot \epsilon$, one can prove that the RE operation differentiates the embedding with motif-induced information:

$$\|Pow(M_{RE}, w_{RE})_u \cdot H_u - Pow(M_{RE}, w_{RE})_v \cdot H_v\|_2$$

$$= \|Pow(M_{RE}, w_{RE})_u \cdot (H_u - H_v) + (Pow(M_{RE}, w_{RE})_u - Pow(M_{RE}, w_{RE})_v) \cdot H_v\|_2$$

$$\geq -\|Pow(M_{RE}, w_{RE})_u \cdot (H_u - H_v)\|_2 + \|(Pow(M_{RE}, w_{RE})_u - Pow(M_{RE}, w_{RE})_v) \cdot H_v\|_2$$

$$\geq -c_2 \cdot \epsilon + (1 + c_2) \cdot \epsilon = \epsilon.$$

The proof is complete.    □

## A.2    PROOF FOR PROPOSITION 1

**Proposition 1.** *RC operation is benefical to stabilzing the model training.*

*Proof.* The RC operation is formulated as

$$RC : H_{RC} = H + w_{RC} \odot H_{SA} \cdot (M_{RC}\mathbb{1}_d^T), \tag{6}$$

where and $\mathbb{1}_d$ is $d$-dimensional all-one column vector. Here $H_{SA}$ introduces the current graph's instance-specific information, i.e., mean representations in each graph. $w_{RC}$ is a learnable weight balancing mini-batch and instance-specific statistics. Assume the number of nodes in each graph is consistent. The expectation of input features after RC, i.e., $\mathbb{E}(H_{RC})$, can be represented as

$$\mathbb{E}(H_{RC}) = (1 + w_{RC} \odot (M_{RC}\mathbb{1}_d^T)) \cdot \mathbb{E}(H). \tag{7}$$

Let us respectively consider the following centering operation of normalization for the original input H and the feature matrix $H_{RC}$ after RC operation,

$$H_{Center-In} = H - \mathbb{E}(H),$$
$$H_{Center-RC} = H_{RC} - \mathbb{E}(H_{RC}), \tag{8}$$

where $H_{Center-In}$ and $H_{Center-RC}$ denote the centering operation on H and $H_{RC}$. To compare the difference between these two centralized features, we perform

$$H_{Center-RC} - H_{Center-In}$$
$$= (H_{RC} - \mathbb{E}(H_{RC})) - (H - \mathbb{E}(H))$$
$$= H + w_{RC} \odot (M_{RC}\mathbb{1}_d^T) \cdot H_{SA} - (1 + w_{RC} \odot (M_{RC}\mathbb{1}_d^T)) \cdot \mathbb{E}(H) - (H - \mathbb{E}(H)) \tag{9}$$
$$= w_{RC} \odot (M_{RC}\mathbb{1}_d^T) \cdot (H_{SA} - \mathbb{E}(H)),$$

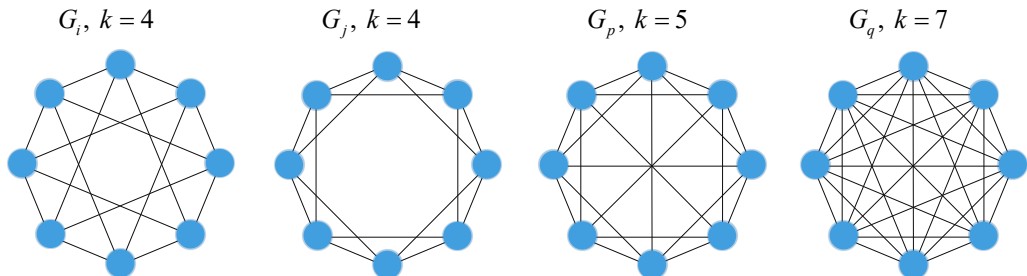

Figure 6: The illustration of four $k$-regular graphs with the same nodes but different structures. When directly performing the summation-normalization operation on motif-induced information, all weights will be equal to 1/8.

where the values in $\mathrm{M_{RC}}$ are always positive numbers. When values in $w_{\mathrm{RC}}$ are close to zero, the centering operation still relies on running statistics over the training set. On the other hand, the importance of graph instance-specific statistics grows when the absolute value of $w_{\mathrm{RC}}$ becomes larger. Here, we ignore the affine transformation operation and assume values larger than the running mean, kept after the following activation layer, are important information for representations, and vice versa. In case of $w_{\mathrm{RC}} > 0$, while $\mathrm{H_{SA}} > \mathbb{E}(\mathrm{H})$, more important information tends to be preserved, and vice versa. In case of $w_{\mathrm{RC}} < 0$, while $\mathrm{H_{SA}} > \mathbb{E}(\mathrm{H})$, the noisy features tend to be weakened, and vice versa. A similar analysis in BatchNorm2D has been provided in (Gao et al., 2021), interested readers please refer to that for details. The main difference is that MotifNorm aims to embed structural information to compensate for ignored characteristics in node-include subgraphs, while RBN is proposed to address the distribution differences.

The proof is complete. □

### A.3 DESIGN OF THE REPRESENTATION CALIBRATION FACTOR

Here, we talk about the design of representation calibration factor $\mathrm{M_{RC}} = \mathrm{M_{SN}} \cdot \mathrm{M_{\mathcal{G}}}$, which is a normalization for motif-induced weights $\mathrm{M_{\mathcal{G}}}$. If we directly adopt the original weights, existing many unequal large values, it will make training oscillating. Thus, the normalization is essential for $\mathrm{M_{\mathcal{G}}}$. However, if we just perform summation-normalization in an arbitrary graph (i.e., $\mathrm{M_{SN}}$), it will not distinguish two graphs with the same nodes but different degrees, e.g., four graphs in Figure 6 where each weight will become 1/8. To this end, we design the above normalization technique to strengthen the motif power for the representation calibration operation.

# B    EXPERIMENTAL DETAILS

## B.1    MORE DETAILS OF BENCHMARK DATASETS AND BASELINE METHODS

**Benchmark Datasets.** For the task of graph property prediction, we select IMDB-BINARY, ogbg-moltoxcast, ogbg-molhiv and ZINC datasets. The IMDB-BINARY is a $k$-regular dataset for binary classification task, which means that each node has a degree $k$. The ogbg-moltoxcast is collected for multi-task classification task, where the number of the tasks is 617. The ogbg-molhiv is a molecule dataset that used for binary classification task, but the ouput dimension of the end-to-end GNNs is 1 because its metric is ROC-AUC. The ZINC is the real-world molecule dataset for the example reconstruction. In this paper, we follow the work in (Dwivedi et al., 2022a) to use ZINC for the task of graph regression. These graph prediction datasets are from (Morris et al., 2020; Hu et al., 2020; Irwin et al., 2012) respectively. For the node level prediction, we select four benchmark datasets including Cora, Pubmed and ogbn-proteins. The first three datasets are the social network and the last one is a protein-protein association network dataset. For the evalutation of link property prediction, we select ogbl-collab dataset in this paper. These node and link prediction datasets are from (Kipf & Welling, 2017; Hu et al., 2020) respectively. More detailed dataset information is provided in Table 8.

**Baseline Methods.** To evalute our proposed MotifNorm module, we need to compare other normalization methods adopted in GNNs for various graph tasks, including BatchNorm (Ioffe & Szegedy, 2015), GraphNorm (Cai et al., 2021), ExpreNorm (Dwivedi et al., 2022a) for graph property prediction, and PairNorm (Zhao & Akoglu, 2020a), NodeNorm (Zhou et al., 2021), MeanNorm (Yang et al., 2020a), GroupNorm (Zhou et al., 2020) for node and link property prediction. A part of these normalization methods are provided in (Chen et al., 2022b), and other source codes are provided by authors. For the backbone GNNs, we consider the most popuare message-passing architectures such as GCN (Kipf & Welling, 2017), GAT (Veličković et al., 2018), GIN (Xu et al., 2019), SGC (Wu et al., 2019) and GraphSage (Hamilton et al., 2017). Specially, we will compare our MotifNorm with all above normlization modules. For the network architectures, we follow CNA architecture, i.e., convolution, normalization and activation. In this paper, we do not adopt any skills like dropedge (Huang et al., 2020; Rong et al., 2020), residual connection (Xu et al., 2018; Li et al., 2019; Liu et al., 2020), etc.

**Experiment Setting.** For different datasets, we provide more detailed statistics information in Table 8. The embedding dimension in each hidden layer on all datasets is set as 128. We optim the GNNs' architectures using torch.optim.lr_scheduler.ReduceLROnPlateau by setting patience step as 10 or 15 to reduce learning rate. The learning rate is $1e-3$ for graph classification, and $1e-2$ for node, link predictions. When the learning rate reduces to $1e-5$, the training will be terminated. More detailed statistics of experimental settings are provided in Table 9. Specially, we split the IMDB-BINARY dataset into train-vallid-test format using a hierarchical architecture. In details, we first segment this dataset according to the edge density information into 10 set (i.e., the edge density $\in \{0.0-0.1\}, \cup, \{0.1-0.2\}..., \{0.9-1.0\}$, and then sort the graphs using the average degree information. Finally, we select the samples in each segment using a fix step size as valid and test samples. The statistic information for splitting the valid and test set of Label-0 and Label-1 on IMDB-BINARY is provide in Table 10. By adopting this splitting scheme, distribution differences among train, valid and test sets are weakened (Experiment results show this contribution fact but without a theoretical basis now). The splitting details are implemented in the datasets/dgl_imdb_dataset.py. To reproduce the comparison results using a single layer of MLP and GIN, the dropout is set to 0.0 and warming up the learning rate from 0.0 to $1e-3$ at the first 50 epoches. When layer is equal to 4, the doupout at the input layer is selected in $\{0.3, 0.4, 0.5\}$ (i.e., -init_dp in the code), and hidden layer is set to 0.5. To draw the Figure 2, we remove the warmup operation for learning rate (i.e., remove –lr_warmup in the shell files).

**Implementation.** MotifNorm needs to process the motif-induced weight into datasets and then load this processing information to embed structural information into node representations. Especially for the node-relevant classification, node representations need to contain the same power before MotifNorm. Thus, we perform the $l$-2 normalization to ensure their power are consistent. The two scripts are at: datasets/preprocess.py and modules/norm/motifnorm.py.

Table 8: The statistic information of 8 benchmark datasets.

| Dataset Name | Dataset Type | Task Type | #Graphs | Avg. #Nodes | Avg. #Edges |
|---|---|---|---|---|---|
| IMDB-BINARY | molecular | Graph classification | 1,000 | 19.8 | 193.1 |
| ogbg-toxcast | molecular | Graph classification | 8,576 | 18.8 | 19.3 |
| ogbg-molhiv | molecular | Graph classification | 41,127 | 25.5 | 27.5 |
| ZINC | molecular | Graph regression | 10,000 | 23.2 | 49.8 |
| Cora | social | Node classification | 1 | 2,708 | 5,429 |
| Pubmed | social | Node classification | 1 | 19,717 | 44,338 |
| ogbn-proteins | proteins | Node classification | 1 | 132,534 | 39,561,252 |
| ogbl-collab | social | Link classification | 1 | 235,868 | 1,285,465 |

Table 9: The detailed experimental settings of GNNs on various graph-structured tasks.

| Name | Metrics | Edge Conv. | Layers | LR | Batch Size | InitDim. | HiDim. | Weight decay | Dropout |
|---|---|---|---|---|---|---|---|---|---|
| IMDB-BINARY | ROC-AUC | False | 1, 4 | $1e-3$ | 32 | 128 | 128 | 0.0 | 0.0, 0.5 |
| ogbg-toxcast | ROC-AUC | False | 4, 16, 32 | $1e-3$ | 128 | 9 | 128 | 0.0 | 0.5 |
| ogbg-molhiv | ROC-AUC | False | 4, 16, 32 | $1e-3$ | 256 | 9 | 128 | 0.0 | 0.5 |
| ZINC | MAE | False | 4, 16, 32 | $1e-3$ | 128 | 1 | 128 | 0.0 | 0.5 |
| Cora | Accuracy | False | $[0;2;32]$ | $1e-2$ | –– | 1433 | 128 | 0.0 | 0.5 |
| Pubmed | Accuracy | False | 4, 16, 32 | $1e-2$ | –– | 500 | 128 | 0.0 | 0.5 |
| ogbn-proteins | ROC-AUC | False | 4, 16, 32 | $1e-2$ | –– | 8 | 128 | 0.0 | 0.5 |
| ogbl-collab | Hits@50 | False | 4, 16, 32 | $1e-2$ | $64 \times 1024$ | 128 | 128 | 0.0 | 0.0 |

Table 10: The statistic information for splitting IMDB-BINARY.

| | 0.0−0.1 | 0.1−0.2 | 0.2−0.3 | 0.3−0.4 | 0.4−0.5 | 0.5−0.6 | 0.6−0.7 | 0.7−0.8 | 0.8−0.9 | 0.9−1.0 | 1.0 | Total Num. |
|---|---|---|---|---|---|---|---|---|---|---|---|---|
| Label-0 | 1 | 17 | 60 | 89 | 49 | 117 | 52 | 12 | 15 | 7 | 81 | 500 |
| Label-1 | 0 | 22 | 81 | 145 | 58 | 97 | 30 | 8 | 1 | 0 | 58 | 500 |
| Total Label | 1 | 39 | 141 | 234 | 107 | 214 | 82 | 20 | 16 | 7 | 139 | 1000 |
| Steps | - | 9 | 8 | 7 | 6 | 5 | 4 | 3 | 2 | 2 | 6 | |
| Label-0 Sel. | 0 | 2 | 14 | 28 | 16 | 46 | 30 | 8 | 14 | 6 | 26 | 190 |
| Label-1 Sel. | 0 | 4 | 20 | 40 | 18 | 38 | 14 | 4 | 0 | 0 | 18 | 156 |
| Total Sel. | 0 | 6 | 34 | 68 | 34 | 84 | 44 | 12 | 14 | 6 | 44 | 346 |

## B.2 MORE COMPARISONS ON GRAPH ISOMORPHISM TEST

Table 11: Experimental results on IMDB-BINARY dataset with various k-regular graphs. The best results under different backbones are highlighted with **boldface.**

| | Normalization | | | L | ROC-AUC | | | LOSS | | |
|---|---|---|---|---|---|---|---|---|---|---|
| | NoNorm | BatchNorm | MotifNorm | | Test Split | Valid Split | Train Split | Test Split | Valid Split | Train Split |
| MLP | ✓ | | | 1 | 56.66 ± 1.09 | 57.05 ± 0.81 | 50.40 ± 0.04 | 0.6961 | 0.6961 | 0.6927 |
| | | ✓ | | 1 | 56.86 ± 0.72 | 57.83 ± 0.84 | 51.38 ± 2.69 | 0.6954 | 0.6954 | 0.6923 |
| | | | ✓ | 1 | **78.16 ± 0.52** | **78.69 ± 0.44** | **79.05 ± 0.34** | 0.5631 | 0.5581 | 0.5519 |
| GIN | ✓ | | | 1 | 77.40 ± 0.20 | 77.69 ± 0.13 | 78.95 ± 0.16 | 0.5684 | 0.5658 | 0.5525 |
| | | ✓ | | 1 | 77.71 ± 0.19 | 78.11 ± 0.10 | 79.65 ± 0.23 | 0.5655 | 0.5593 | 0.5391 |
| | | | ✓ | 1 | **78.42 ± 0.15** | **78.89 ± 0.15** | **80.24 ± 0.18** | 0.5584 | 0.5565 | 0.5281 |
| GSN | | ✓ | | 1 | 77.50 ± 0.18 | 77.54 ± 0.16 | 78.67 ± 0.17 | 0.5696 | 0.5688 | 0.5570 |
| GraphSNN | | ✓ | | 1 | 77.52 ± 0.16 | 77.61 ± 0.18 | 78.90 ± 0.15 | 0.5691 | 0.5671 | 0.5520 |
| GIN | ✓ | | | 4 | 78.22 ± 0.41 | 78.27 ± 0.45 | 80.03 ± 0.30 | 0.5602 | 0.5553 | 0.5304 |
| | | ✓ | | 4 | 78.41 ± 0.21 | 78.76 ± 0.17 | 80.10 ± 0.14 | 0.5625 | 0.5590 | 0.5325 |
| | | | ✓ | 4 | **79.55 ± 0.35** | **79.62 ± 0.38** | **81.50 ± 0.48** | 0.5539 | 0.5510 | 0.5133 |
| GCN | ✓ | | | 4 | 68.99 ± 1.38 | 69.08 ± 1.81 | 68.09 ± 1.81 | 0.6920 | 0.6921 | 0.6850 |
| | | ✓ | | 4 | 76.75 ± 1.31 | 76.96 ± 0.69 | 75.70 ± 1.05 | 0.5755 | 0.5640 | 0.5838 |
| | | | ✓ | 4 | **78.78 ± 1.01** | **79.03 ± 0.61** | **78.69 ± 0.84** | 0.5597 | 0.5504 | 0.5493 |
| GAT | ✓ | | | 4 | 69.07 ± 1.59 | 69.78 ± 1.65 | 66.78 ± 1.68 | 0.6904 | 0.6891 | 0.6887 |
| | | ✓ | | 4 | 75.10 ± 1.51 | 75.95 ± 1.27 | 75.26 ± 0.87 | 0.5866 | 0.5852 | 0.5978 |
| | | | ✓ | 4 | **78.87 ± 0.80** | **79.20 ± 0.56** | **79.19 ± 0.78** | 0.5559 | 0.5448 | 0.5504 |
| GraphSage | ✓ | | | 4 | 64.35 ± 4.36 | 65.97 ± 4.85 | 63.49 ± 2.93 | 0.6923 | 0.6921 | 0.6918 |
| | | ✓ | | 4 | 75.06 ± 1.77 | 76.12 ± 1.31 | 75.95 ± 1.05 | 0.5819 | 0.5748 | 0.5805 |
| | | | ✓ | 4 | **78.93 ± 0.99** | **79.02 ± 0.64** | **79.23 ± 0.49** | 0.5534 | 0.5481 | 0.5447 |
| SGC | ✓ | | | 4 | 62.83 ± 2.66 | 66.19 ± 2.86 | 62.30 ± 2.37 | 0.6951 | 0.6941 | 0.6876 |
| | | ✓ | | 4 | 70.94 ± 1.33 | 73.51 ± 1.02 | 71.20 ± 0.65 | 0.6231 | 0.6044 | 0.6109 |
| | | | ✓ | 4 | **78.85 ± 0.70** | **78.53 ± 0.83** | **78.61 ± 0.63** | 0.5577 | 0.5556 | 0.5466 |
| GSN | | ✓ | | 4 | 78.90 ± 0.63 | 79.28 ± 0.70 | 80.70 ± 0.37 | 0.5555 | 0.5543 | 0.5377 |
| GraphSNN | | ✓ | | 4 | 79.16 ± 0.67 | 79.35 ± 0.82 | 80.74 ± 0.44 | 0.5541 | 0.5530 | 0.5356 |

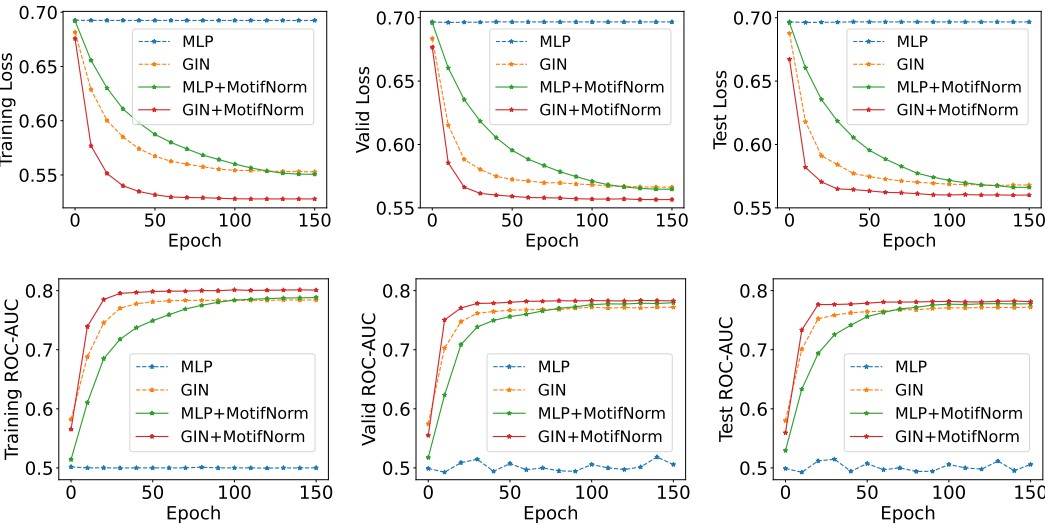

Figure 7: Learning curves of one-layer MLP, GIN, MLP + MotifNorm and GIN + MotifNorm on IMDB-BINARY dataset (without learning rate warmup).

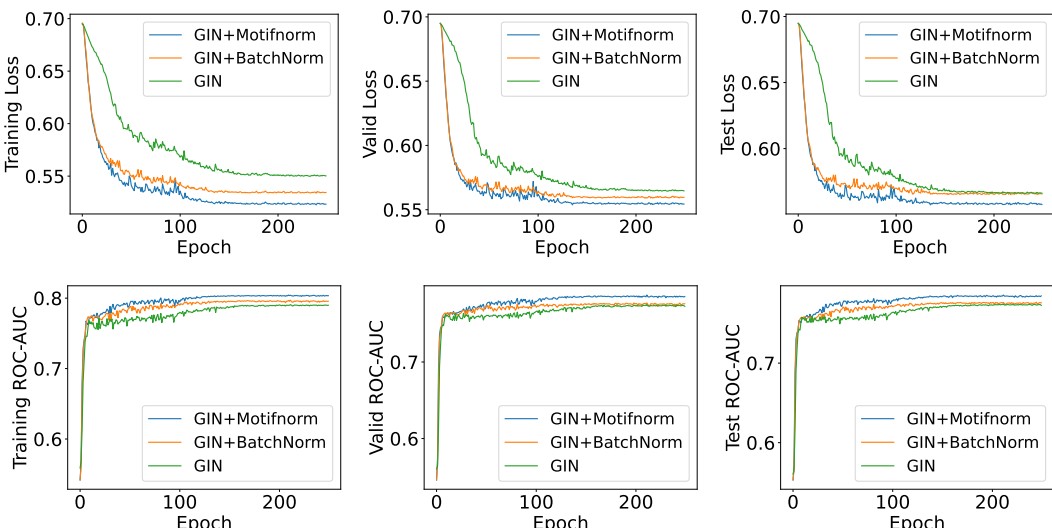

Figure 8: Learning curves of one-layer GIN, GIN+BatchNorm and GIN+MotifNorm on IMDB-BINARY dataset with learning rate warmup (i.e., the curves of the reported scores in Table 2).

### B.3 MORE COMPARISONS ON THE OVER-SMOOTHING ISSUE

This section provide the illustration of `GCN` (Kipf & Welling, 2017) and `GraphSage` (Hamilton et al., 2017) with different six existent normalizations on `Cora` dataset.

Given the node embedding $X = [x_1, x_2, ..., x_n]^T \in \mathbb{R}^{n \times d}$, where $x_i \in \mathbb{R}^d$ denotes the node embedding at layer $i$. The existing normalization methods for GNNs include PairNorm (Zhao & Akoglu, 2020a), NodeNorm (Zhou et al., 2021), MeanNorm (Yang et al., 2020a), GroupNorm (Zhou et al., 2020) and BatchNorm (Ioffe & Szegedy, 2015) are depicted as follows:

PairNorm (Zhao & Akoglu, 2020a).

$$
\begin{aligned}
\tilde{x}_i &= x_i - \frac{1}{n} \sum_{i=1}^{n} x_i, \\
\texttt{PairNorm}(x_i, s) &= \frac{s \cdot \tilde{x}_i}{\sqrt{\frac{1}{n} \sum_{i=1}^{n} ||\tilde{x}_i||_2^2}}.
\end{aligned}
\tag{10}
$$

NodeNorm (Zhou et al., 2021).

$$
\texttt{NodeNorm}(x_i, p) = \frac{x_i}{\text{std}(x_i)^{\frac{1}{p}}}.
\tag{11}
$$

MeanNorm (Yang et al., 2020a).

$$
\texttt{MeanNorm}(x_{(k)}) = x_{(k)} - \mathbb{E}[x_{(k)}].
\tag{12}
$$

BatchNorm (Ioffe & Szegedy, 2015).

$$
\texttt{BatchNorm}(X) = \frac{X - \mathbb{E}(X)}{\sqrt{\mathbb{D}(X) + \epsilon}} \odot \gamma + \beta.
\tag{13}
$$

GroupNorm (Zhou et al., 2020).

$$
\begin{aligned}
\texttt{GroupNorm}(X; G, \lambda) &= X + \lambda \cdot \sum_{g=1}^{G} \texttt{BatchNorm}(\tilde{X}_g), \\
where \ \tilde{X}_g &= \text{softmax}(X \cdot U)[:, g] \circ X.
\end{aligned}
\tag{14}
$$

MotifNorm please refer to Eq. (5) for details.

Here $x_{(i)} \in \mathbb{R}^n$ denotes the $i$-th column of X, $s$ in PairNorm is a hyperparameter controlling the average pair-wise variance and we choose $s = 1$ in our case. $p$ in NodeNorm denotes the normalization order and our paper uses $p = 2$.

The t-SNE illustrations of different normalizations embedded in GCN and GraphSage are provided in Figure 9 and 10 respectively, where the number of layers are 4, 16, 32. Figure 11 shows the over-smoothing phenomenon with the layers increasing using GraphSage, where the number of layers is from 16 to 32 with the step size as 2.

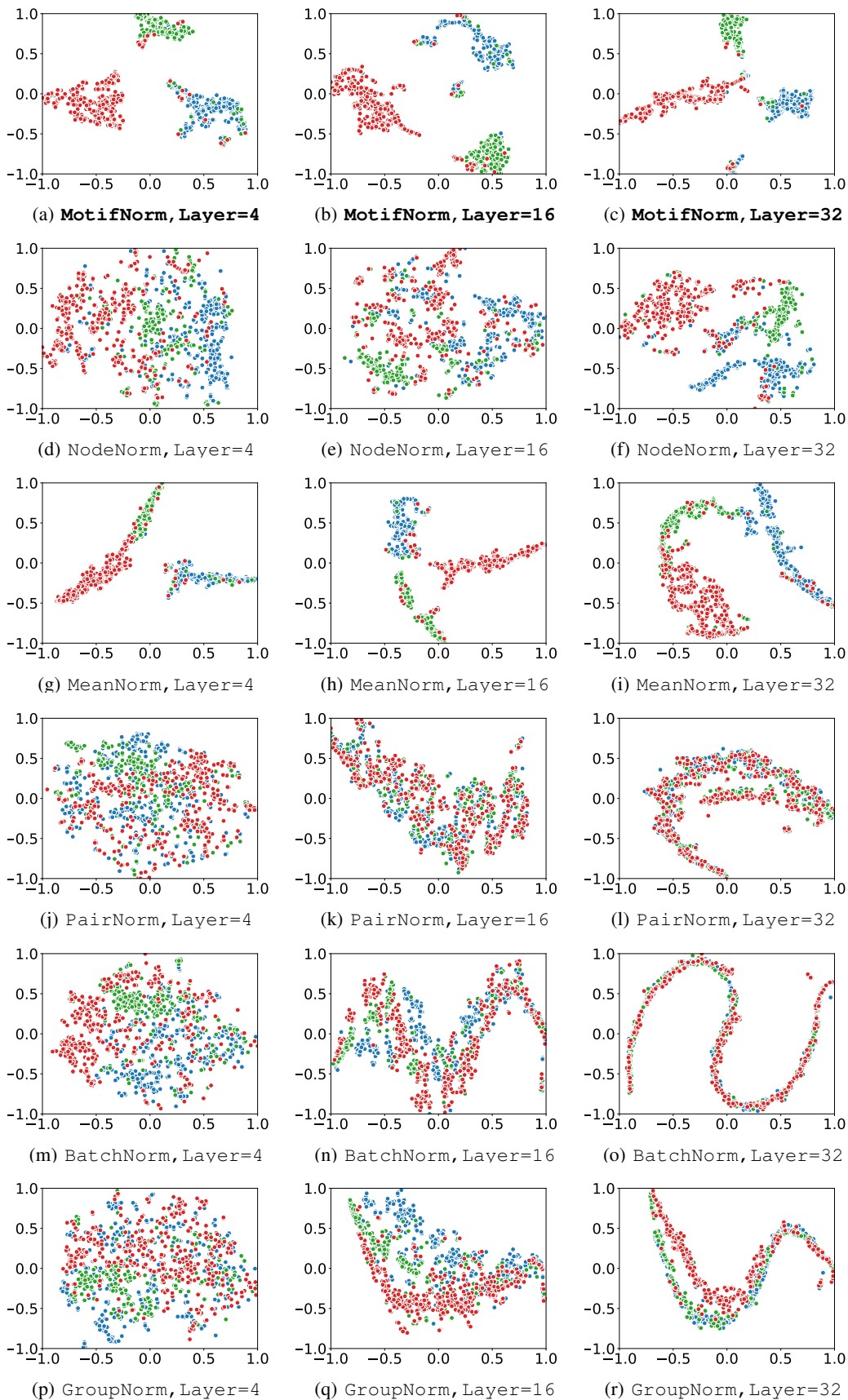

Figure 9: The t-SNE visualization using GCN with different normalization methods on Cora dataset.

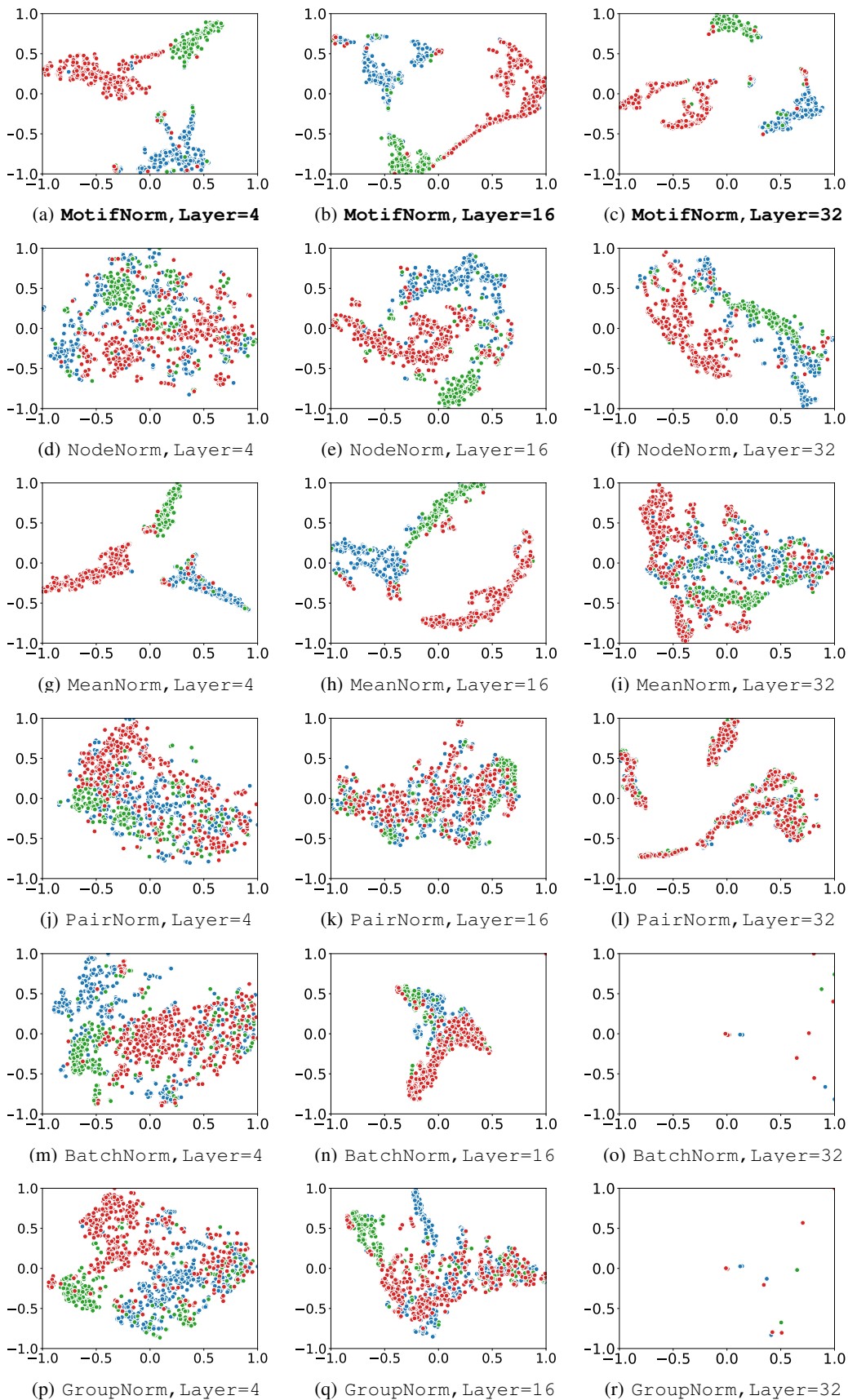

Figure 10: The t-SNE visualization using GraphSage with different normalization methods on Cora dataset.

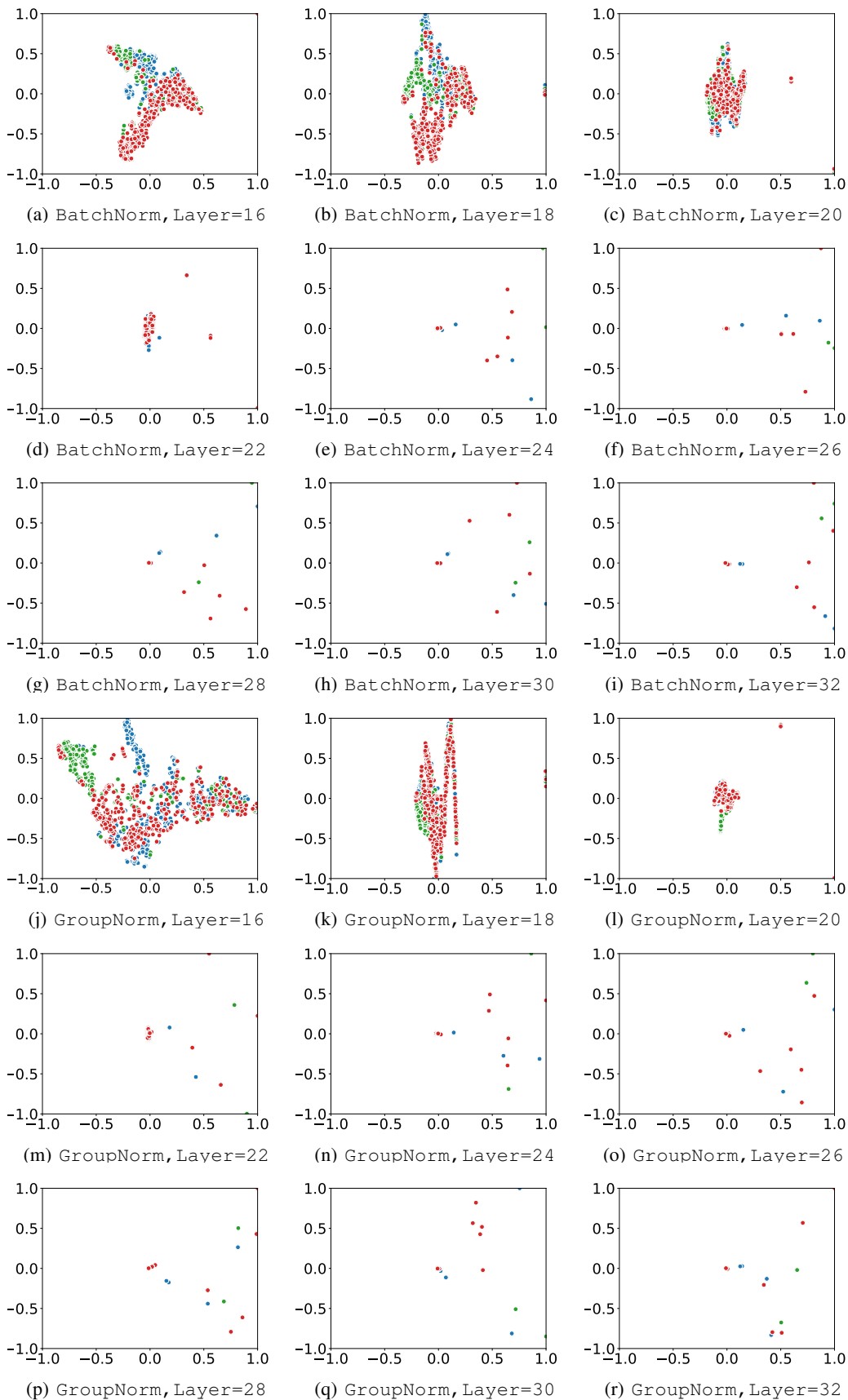

Figure 11: The t-SNE visualization using GraphSage with BatchNorm and GroupNorm on Cora dataset.

### B.4 MORE STATISTICS ON THE OTHER SIX DATASETS

Table 12: Experimental results of different normalization methods on the graph prediction task. We use GCN, GAT and GIN as the basic backbones and set the number of layers as 4, 16 and 32. The best results on each dataset are highlighted with **boldface.**

| | Methods | ogbg-moltoxcast | | | ogbg-molhiv | | | ZINC | | |
| | | 4 | 16 | 32 | 4 | 16 | 32 | 4 | 16 | 32 |
|---|---|---|---|---|---|---|---|---|---|---|
| GCN | NoNorm | 61.13 | 59.34 | 56.08 | 76.01 | 71.90 | 60.59 | 0.643 | 0.690 | 0.748 |
| | PairNorm | 60.69 | 59.45 | 55.07 | 74.06 | 72.13 | 62.25 | 0.573 | 0.569 | 0.597 |
| | NodeNorm | 61.94 | 55.58 | 49.90 | 74.80 | 57.75 | 57.64 | 0.625 | 1.332 | 1.547 |
| | MeanNorm | 63.21 | 60.42 | 54.76 | 74.42 | 72.39 | 60.59 | 0.602 | 0.637 | 0.695 |
| | GroupNorm | 61.58 | 59.48 | 56.73 | 76.66 | 71.84 | 66.38 | 0.641 | 0.673 | 0.737 |
| | GraphNorm | 60.78 | 53.75 | 53.36 | 75.59 | 65.55 | 66.49 | 0.592 | 0.655 | 1.547 |
| | BatchNorm | 63.39 | 59.73 | 53.47 | 76.11 | 76.62 | 74.21 | 0.573 | 0.611 | 0.655 |
| | ExpreNorm | 64.97 | 57.91 | 57.82 | 76.05 | 76.75 | 72.36 | 0.564 | 0.570 | 0.646 |
| | **MotifNorm** | **66.92** | **65.19** | **63.40** | **77.29** | **77.71** | **75.99** | **0.489** | **0.524** | **0.523** |
| GAT | NoNorm | 62.61 | 50.84 | 50.12 | 76.71 | 57.38 | 50.64 | 0.714 | 1.541 | 1.547 |
| | GraphNorm | 60.53 | 52.79 | 53.22 | 75.30 | 73.86 | 64.03 | 0.576 | 1.254 | 1.537 |
| | BatchNorm | 63.31 | 53.39 | 53.24 | 76.07 | 76.87 | 73.74 | 0.585 | 0.624 | 0.643 |
| | ExpreNorm | 65.56 | 57.65 | 57.60 | 76.99 | 72.24 | 72.56 | 0.555 | 0.562 | 1.451 |
| | **MotifNorm** | **66.57** | **64.04** | **58.26** | **77.36** | **77.08** | **76.70** | **0.495** | **0.517** | **0.522** |
| GIN | NoNorm | 62.19 | 56.38 | 54.83 | 76.33 | 69.70 | 58.87 | 0.496 | 0.520 | 1.069 |
| | GraphNorm | 62.44 | 54.95 | 55.72 | 76.55 | 66.00 | 67.01 | 0.462 | 1.203 | 1.446 |
| | BatchNorm | 63.72 | 58.67 | 55.56 | 76.62 | 70.28 | 66.82 | 0.477 | 0.516 | 1.153 |
| | ExpreNorm | 65.98 | 57.80 | 56.56 | 76.23 | 69.97 | 70.96 | 0.438 | 0.482 | 1.157 |
| | **MotifNorm** | **66.65** | **63.01** | **57.46** | **77.38** | **73.03** | **72.89** | **0.410** | **0.458** | **0.902** |

Table 13: Experimental results of different normalization methods on the node prediction task and link prediction task. We use GCN, and GraphSage as the basic backbones and set the number of layers as 4, 16 and 32. The best results on each dataset are highlighted with **boldface.**

| | Settings | Pubmed | | | ogbn-proteins | | | ogbl-collab | | |
| | | 4 | 16 | 32 | 4 | 16 | 32 | 4 | 16 | 32 |
|---|---|---|---|---|---|---|---|---|---|---|
| GCN | NoNorm | 76.16 | 54.67 | 45.58 | 69.16 | 63.24 | 63.15 | 35.38 | 22.11 | 15.24 |
| | PairNorm | 74.25 | 56.24 | 55.13 | 69.28 | 63.15 | 63.00 | 31.26 | 23.22 | 14.69 |
| | NodeNorm | 76.02 | 40.87 | 41.18 | 70.17 | 63.50 | 63.23 | 27.48 | 08.48 | 08.28 |
| | MeanNorm | 76.05 | 73.40 | 65.34 | 69.14 | 63.05 | 62.40 | 33.28 | 22.56 | 16.16 |
| | GroupNorm | 76.19 | 63.55 | 54.84 | 70.25 | 62.74 | 63.63 | 35.28 | 27.41 | 20.27 |
| | BatchNorm | 75.62 | 48.88 | 43.28 | 69.96 | 67.36 | 63.86 | 47.57 | 26.14 | 21.68 |
| | **MotifNorm** | **77.08** | **76.66** | **67.81** | **71.69** | **68.66** | **68.05** | **51.65** | **50.01** | **47.65** |
| GraphSage | NoNorm | 76.94 | 40.65 | 41.67 | 66.05 | 60.56 | 60.47 | 25.27 | 02.08 | 00.00 |
| | PairNorm | 72.78 | 53.02 | 45.90 | 62.29 | 60.53 | 60.32 | 41.72 | 16.88 | 12.44 |
| | NodeNorm | 77.22 | 40.64 | 40.64 | 64.48 | 62.63 | 61.89 | 19.74 | 02.57 | 02.62 |
| | MeanNorm | 76.68 | 58.70 | 47.48 | 63.69 | 61.03 | 52.06 | 46.17 | 21.54 | 13.16 |
| | GroupNorm | 76.83 | 40.42 | 43.49 | 68.09 | 61.58 | 60.60 | 45.43 | 23.98 | 15.43 |
| | BatchNorm | 75.49 | 45.11 | 42.74 | 63.75 | 62.96 | 61.54 | 47.05 | 23.01 | 14.89 |
| | **MotifNorm** | **77.48** | **76.54** | **73.68** | **67.81** | **67.02** | **66.07** | **52.31** | **48.94** | **48.39** |

