# OpenReview forum: "Motif-induced Graph Normalization"
_ICLR.cc/2023/Conference — Submitted to ICLR 2023_

### Official Review · Reviewer_WzdD · 2022-10-22

**Confidence:** 4
**Clarity, Quality, Novelty And Reproducibility:** 1. This manuscript is clearly written…
**Correctness:** 3
**Technical Novelty And Significance:** 2
**Empirical Novelty And Significance:** 3
**Recommendation:** 5

**Strength And Weaknesses:**

Strengths

1. MotifNorm, as a plug-in, makes GNNs more expressive (beyond 1-WL test) and less prone to the over-smoothing problem.
2. The calibration and enhancement steps in MotifNorm is novel and efficient.
3. Good experimental results.

Weeknesses

1. This work only provides proof of the expressiveness of GNN+ MotifNorm beyond 1-WL in the case of k-regular graphs. However, there are few k-regular graphs in real applications. The expressiveness analysis will be more convincing if the author could give proof in more general cases.
2. The motif-induced weight is predefined and does not consider any node or edge attributes, or the task for prediction. This limits MotifNorm ability to handle complex scenarios like heterogenous graphs or graphs with edge attributes.
3. Table 2 does not provide the performance of GSM+MotifNorm and GraphSNN+MotifNorm. Since GSM and GraphSNN are convolutional approaches that achieves higher expressivity than 1-WL test, it is desired to see whether MotifNorm will further enhance the expressivity of GSM/ GraphSNN.
4. Figure 5(a): For GCN with shallow layers, GCN+MotifNorm performs worse than GroupNorm and NodeNorm. It seems MotifNorm really helps in alleviating over-smoothing but shows no advantages in shallow models. Could the authors explain a bit?
5. Missing discussions with popular normalization methods. Also, there are some recent related works. For example, [1] proposes a normalization technique with better flexibility. [2] also proposes a normalization approach to alleviate over-smoothing problem. It is better to compare with works with similar motivations and demonstrate the advantage of the proposed approach.

References
[1] Chen, Y., Tang, X., Qi, X., Li, C. G., & Xiao, R. (2022). Learning graph normalization for graph neural networks. Neurocomputing, 493, 613-625.
[2] Liang, Langzhang, et al. "ResNorm: Tackling Long-tailed Degree Distribution Issue in Graph Neural Networks via Normalization." arXiv preprint arXiv:2206.08181 (2022).


**Summary Of The Paper:**

This work looks to enhance the discriminate capabilities of GNNs by adding a normalization module after the convolution layer. Traditional GNNs suffer from limited expressive power, i.e., upper bounded by 1-WL test, and the over-smoothing problem when stacking multiple GNN layers. To tackle the above two challenges, a motif-induced normalization, MotifNorm, is proposed with theoretical analysis showing that MotifNorm make arbitrary GNNs more expressive than the 1-WL test in distinguishing k-regular graphs. It is also shown that MotifNorm is able to reduce the over-smoothing problem. The experiment results on ten benchmark datasets are promising

**Summary Of The Review:**

The idea is novel. But the above problems should be addressed before publication.

---

> ### Author Response · Authors · 2022-11-19
> **Response to reviewer WzdD (Part-2)**
>
>
>
>
> **Q5. Missing discussions with popular normalization methods. Also, there are some recent related works. For example, [1] proposes a normalization technique with better flexibility. [2] also proposes a normalization approach to alleviate over-smoothing problem. It is better to compare with works with similar motivations and demonstrate the advantage of the proposed approach.**
>
> **R5**. Thanks for this suggestion. we provided the introduction and discussion of  these two methods in the **Section Related Work of the revision (Page 2-3,).** Furthermore, we add the **comparison result with [1] in the Table 3 of the revision(Page 8).**  The comparisons with UnityNorm[1] on graph predictions by using GCN as follows:
>
>
>
> |                        | ogbg-moltoxcast |             |             | ogbg-molhiv |             |             | ZINC        |             |             |
> | ---------------------- | --------------- | ----------- | ----------- | ----------- | ----------- | ----------- | ----------- | ----------- | ----------- |
> |                        | layer=4         | layer=16    | layer=32    | layer=4     | layer=16    | layer=32    | layer=4     | layer=16    | layer=32    |
> | GCN+BatchNorm          | 63.39           | 59.73       | 53.47       | 76.11       | 76.62       | 74.21       | 0.573       | 0.611       | 0.655       |
> | GCN+***UnityNorm[1]*** | ***63.86***     | ***61.94*** | ***59.18*** | ***75.94*** | ***72.14*** | ***69.44*** | ***0.552*** | ***0.576*** | ***0.650*** |
> | GCN+**MotifNorm**      | **66.92**       | **65.19**   | **63.40**   | **77.29**   | **77.71**   | **75.99**   | **0.489**   | **0.524**   | **0.523**   |
>
>
>
> **Q6. Minor comment: The experimental settings may not be optimal for some baseline models. For example, in Table 3, the original paper result for GCN+GraphNorm on ogbg-molhiv is 78.30 (num_layer=5), but the reproduced result is 75.59 (num_layer=4). My concern for performance drop is that reproduction may not achieve optimal.**
>
> **R6**. The reason is that the setting is different. To achieve that performance, we need to follow the settings: **covolution with edge features**, input dropout as 0.0,  hidden dim 300, **weight decay in {5e-5, 1e-5}, residual connection**, and achieve the results:
>
> |               | ogbg-molhiv   |
> | ------------- | ------------- |
> | GCN+BatchNorm | 78.07 ± 0.782 |
> | GCN+MotifNorm | 78.76 ± 0.371 |
>
> ***On page 8 of the revision, we updated the comparisons with empirical tricks in Table 5 and Table 6 to further improve the quality of the manuscript.***
>
>
> **Ref.**
>
> [1] Learning graph normalization for graph neural networks. Neurocomputing, 2022.
>
> [2] Resnorm: Tackling long-tailed degree distribution issue in graph neural networks via normalization. Arxiv, 2022.
>
>
>
> **Summary of the response.**
>
> Thanks for your time and comments again.
>
> Accordingly to your comments, we have carefully revised paper and give the above responses.
>
> We would be grateful if the reviewer could reconsider the score based on our clarifications and revisions, or let us know if further clarifications are required.

---

> ### Author Response · Authors · 2022-11-19
> **Response to reviewer WzdD (Part-1)**
>
>
>
> We thank the reviewer for the positive comments on our work.
>
> Next we give the responses to address concerns as follows:
>
> **Q1. This work only provides proof of the expressiveness of GNN+ MotifNorm beyond 1-WL in the case of k-regular graphs. However, there are few k-regular graphs in real applications. The expressiveness analysis will be more convincing if the author could give proof in more general cases.**
>
> **R1**. Thanks for the constructive suggestion.  The commonly used message-passing GNNs are weaker than the 1-WL test, and there are no quantitative analyses to measure their expressiveness gap, which makes the expressiveness analysis between an arbitrary GNN+ MotifNorm and 1-WL test difficult. To address the reviewer's concern, we try to give the expressiveness analysis from the perspective of addressing a more general case, i.e., **subtree-isomorphic**, which is a more general phenomenon in real applications. MotfiNorm compensates for the structural information to distinguish the subtree-isomorphic case that 1-WL can not recognize, and thus generally improve the graph expressiveness. Specially, we give the two cases of GNN+ MotifNorm beyond the 1-WL test (1). GNN+ MotifNorm beyond 1-WL in the case of k-regular graphs. (2) GNN+ MotifNorm will go beyond the 1-WL test in an arbitrary graph classification task when the adopted GNNs are 1-WL-equivalent GNNs. e.g., GIN.
>
> ***On page 4 of the revision,  we updated the above context in Expressivity Analysis.***
>
>
>
> **Q2. The motif-induced weight is predefined and does not consider any node or edge attributes, or the task for prediction. This limits MotifNorm ability to handle complex scenarios like heterogenous graphs or graphs with edge attributes.**
>
> **R2**. MotifNorm works with the GNNs methods to compemsate for the ignored information by GNNs, which means that it is not directly related to the type of the data, i.e.,  our MotifNorm can be used as long as the GNNs can handle this type of graph. For example, the heterogeneous graphs are usually mapped into a latent space, and then use the GNNs Learning the node representations. In this background, MotifNorm just works as a plug-in module to compensate for the node-induced substructural information. Similarly, for the  graphs with edge attributes, MotfiNorm can also enhance the performance, e.g., the comparison results as follows: (**we perform GCN convolution with edge attributes**)
>
> |               | ogbg-molhiv   |
> | ------------- | ------------- |
> | GCN+NoNorm    | 77.21 ± 0.430 |
> | GCN+UnityNorm | 77.56 ± 1.060 |
> | GCN+ExpreNorm | 77.99 ± 0.545 |
> | GCN+GraphNorm | 78.10 ± 1.115 |
> | GCN+BatchNorm | 78.07 ± 0.782 |
> | GCN+MotifNorm | 78.76 ± 0.371 |
>
> ***On page 8 of the revision, we supplied the above results in Table 5.***
>
>
> **Q3. Table 2 does not provide the performance of GSM+MotifNorm and GraphSNN+MotifNorm. Since GSM and GraphSNN are convolutional approaches that achieves higher expressivity than 1-WL test, it is desired to see whether MotifNorm will further enhance the expressivity of GSM/ GraphSNN**
>
> **R3**. Thanks.  MotifNorm can further enhance the expressivity of GSN and GraphSNN as:
>
> |                        | Test Split       | Valid Split      |
> | ---------------------- | ---------------- | ---------------- |
> | GSN+BatchNorm          | 78.90 ± 0.63     | 79.28 ± 0.70     |
> | **GSN+MotifNorm**      | **79.22 ± 0.71** | **79.70 ± 0.59** |
> | GraphSNN+BatchNorm     | 79.16 ± 0.67     | 79.35 ± 0.82     |
> | **GraphSNN+MotifNorm** | **79.89 ± 0.74** | **79.93 ± 0.77** |
>
> ***On page 6 of the revision, we have updated the Table 2***
>
>
> **Q4. Figure 5(a): For GCN with shallow layers, GCN+MotifNorm performs worse than GroupNorm and NodeNorm. It seems MotifNorm really helps in alleviating over-smoothing but shows no advantages in shallow models. Could the authors explain a bit?**
>
> **R4**. We give the explanation as:  The cora is a social network with the input feature dimension 1433. When GCN with shallow layers on cora dataset, the nodes' expressivities are still more correlated with the initial features (i.e., 1433-dimension features). Thus structural-injected MotifNorm may  performs worse than  GroupNorm and NodeNorm, which perform the statistics normalization on Node itself.

---

> ### Author Response · Authors · 2022-11-26
> **Looking forward to your further feedback**
>
> Dear Reviewer WzdD,
>
> We sincerely thank you for the valuable suggestions that have helped us improve the quality of the paper significantly. Our work attempts to design a powerful normalization module to compensate for the sub-structural characteristics ignored by the message-passing GNNs.
>
> We hope our responses could address your concerns.  And we would really appreciate the opportunity to discuss this further if our responses have not already addressed your concerns.
>
> Yours sincerely,
>
> Authors

---

### Official Review · Reviewer_BFQU · 2022-10-22

**Confidence:** 3
**Correctness:** 3
**Technical Novelty And Significance:** 2
**Empirical Novelty And Significance:** 3
**Recommendation:** 6

**Clarity, Quality, Novelty And Reproducibility:**

The paper is clear written. I do not think there is an issue with reproducibility.

**Strength And Weaknesses:**

Strength:
The paper is well written. And the theoretical analysis is also quite clear, which I truly appreciated.
The proposed methods do boosted the performance and the ablation study also revealed the results.

Weakness:
Based on the formulation of "motifnorm", I do not think it is a normalization method. Essentially, it added the information on top of the feature information. And applied the normalization methods. And the definition of "motif" is not quite intuitive. Per my under standing, motif should be some rather stable structure in the graph. But from the method, the motif is equivalent to the subgraph including the node of interest, which makes the method quite similar to the methods like labeling trick (Zhang et al).
I checked the proof of theorem 3. It seems, it does not show how the method will solve the over smooth issue. Actually even there is empirical result, the theorem is not intuitive to me. The norm is also a way to smoothing from the information from the "motif". How would it solve the over smoothing?



**Summary Of The Paper:**

This paper applied a motifnorm to introduce the graph structure into the node presentation and increased the expressive power over 1-WL test. It is well written and with extensive experiments.

**Summary Of The Review:**

This paper is well written and quite complete. But I would have question about the definition of the "norm" and "motif". And the method in some extend shared certain similarity of current methods.

---

> ### Author Response · Authors · 2022-11-19
> **Response to BFQU**
>
>
> We thank the reviewer for the positive comments on our work.
>
> We will give the responses in order.
>
> **Q1. Based on the formulation of "motifnorm", I do not think it is a normalization method. Essentially, it added the information on top of the feature information. And applied the normalization methods. And the definition of "motif" is not quite intuitive. Per my under standing, motif should be some rather stable structure in the graph. But from the method, the motif is equivalent to the subgraph including the node of interest, which makes the method quite similar to the methods like labeling trick (Zhang et al).**
>
> **R1**. We propose the MotifNorm by embedding the structural information into Standard BatchNorm instead of  proposing a whole new framework. Moreover, the proposed MotifNorm contains the basis normalization operations like centering, scaling and affine transformation operations. Motif is stable structure in the **molecular graph**, In this paper, we use its general definition, i.e., node-induced subgraph[1].  Thanks for the recommendation of [2],  which **learns the multi-node representation** for link prediction.  Different from them, we mine the  characteristics in node-include subgraph to inject into Normalization to generally improve the GNNs' expressivity for various tasks.
>
>
>
> **Q2. I checked the proof of theorem 3. It seems, it does not show how the method will solve the over smooth issue. Actually even there is empirical result, the theorem is not intuitive to me. The norm is also a way to smoothing from the information from the "motif". How would it solve the over smoothing?**
>
> **R2**. MotifNorm alleviates the over-smoothing issue by differentiating the embedding via the structural information. Assume that two node representations are similar, and the difference of their node-induced information is greater than a fixed constant, the RE and RC operation can differentiate the embedding. To clear explain that MotifNorm can alleviate the over-smoothing issue, **we have updated the theorem analysis in the revision (Page 4, Theorem 1).**
>
> **Ref.**
>
> [1] Fast neural subgraph matching and counting. Stanford University, CS224W: Machine Learning with Graphs, 2021. URL http://cs224w.stanford.edu/
>
> [2] Labeling trick: A theory of using graph neural networks for multi-node representation learning. NeurIPS, 2021.
>
>
>
> **Summary of the response.**
>
> Thanks for your time and comments again.
>
> Accordingly to your comments, we have carefully revised paper and give the above responses.
>
> We hope our responses can solve your confusion, or let us know if further clarifications are required.

---

> ### Author Response · Authors · 2022-11-26
> **Looking forward to your further feedback**
>
> Dear Reviewer BFQU,
>
> We sincerely thank you for the valuable suggestions that have helped us improve the quality of the paper significantly. Our work attempts to design a powerful normalization module to compensate for the sub-structural characteristics ignored by the message-passing GNNs.
>
> We hope our responses could address your concerns.  And we would really appreciate the opportunity to discuss this further if our responses have not already addressed your concerns.
>
> Yours sincerely,
>
> Authors

---

### Official Review · Reviewer_iJ89 · 2022-10-23

**Confidence:** 4
**Correctness:** 2
**Technical Novelty And Significance:** 2
**Empirical Novelty And Significance:** 2
**Recommendation:** 3

**Clarity, Quality, Novelty And Reproducibility:**

As I described above, I found the paper has limited novelty and weak motivation. Besides, the correctness of theoretical analyses and confidence of empirical results are also doubtful.

**Strength And Weaknesses:**

### Strength

- Normalization methods are known to help the training of deep neural networks and it is an important direction to design normalization methods for GNNs.

- It is reasonable to conduct comprehensive experiments covering node/link/graph-level tasks to examine the proposed MotifNorm for different GNN variants.

### Weaknesses
Here are several critical points that need to be essentially addressed.

**[The motivation and novelty of the proposed MotifNorm]**

- The motivation is not strong. The authors state that the designed weight factor needs at least to distinguish two subgraphs shown in Figure 2 (in page 3). However, it is not general to just use one pair of graphs as measurement. Such motivation is not convincing to support the following network design.

- The novelty of the proposed MotifNorm is limited. The core step Representation Calibration (RC) of the proposed MotifNorm is highly related to the Representative Batch Normalization (RBN) [1], and the proof of Theorem 2 mainly follows [1]. The authors should carefully discuss the relation between RBN and MotifNorm.

**[The correctness of theoretical analyses]**

- Definitions and theorems presented in this paper are not rigorously written. In Definition 1, there do not exist definitions of the symbol $u$ and $u'$. In Theorem 1, readers would be confused about which graphs the subgraphs $S_{v_i}$ and $S_{v_j}$ belong to (Please refer to [2] for formal statements). Moreover, both Theorem 2 and Theorem 3 are not quantitative analyses, which deliver little insightful information on the proposed MotifNorm.

- There exist mistakes in proofs for these theorems. For example, in Appendix A.2, the authors remove the $\mathbf{M}$$_{RC}$ term to analyze the effect of representation calibration, however, it is non-trivial to conclude the final statements with the RC term. In Appendix A.3, the analysis on the over-smoothing issues is based on the equation 14, but this form largely differs from detailed implementations of the proposed MotifNorm. Thus, the correctness of theoretical analyses in this paper is doubtful.

**[The confidence of experimental results]**:

- The results of baselines on several datasets are relatively lower than results in previous works. For example, for ZINC dataset, GCN with BatchNorm achieve 0.416 and 0.278 MAE respectively [3]. For ogbg-molhiv dataset, GCN with GraphNorm achieves 78.30 ROC-AUC [4]. For ogbn-proteins dataset, GCN with BatchNorm achieves 0.7251 ROC-AUC [3]. For ogbn-collab dataset, GraphSAGE with BatchNorm achieves 54.63% Hits@50 [3]. These results from previous works outperform the best results reported in this paper. Thus, the experimental results are doubtful and need to be clarified. Furthermore, many baselines are out-of-date and more recent methods such as  graph transformers should be considered as baselines [If the proposed MotifNorm cannot bring SoTA results, why should people use it?].

- Runtime/Memory Cost evaluation should be provided for completeness. In equation 3, the structural weight need to count the number of edges and nodes in the node-induced subgraph, which increase additional computational costs for standard normalization methods. It is better to provide runtime and memory cost evaluation for comparison. Besides, the authors report the number of epochs that each model take to achieve the corresponding performance in Table 2. We can see that GNNs with MotifNorm converges slower than GNNs with BatchNorm in some cases. Thus, it is also reasonable to provide the total time that GNNs with different normalization methods take to achieve the reported scores.

[1] Gao, Shang-Hua, et al. "Representative batch normalization with feature calibration." Proceedings of the IEEE/CVF Conference on Computer Vision and Pattern Recognition. 2021.

[2] Wijesinghe, Asiri, and Qing Wang. "A New Perspective on" How Graph Neural Networks Go Beyond Weisfeiler-Lehman?"." International Conference on Learning Representations. 2022.

[3] Dwivedi, Vijay Prakash, et al. "Benchmarking graph neural networks." arXiv preprint arXiv:2003.00982 (2020).

[4] Cai, Tianle, et al. "Graphnorm: A principled approach to accelerating graph neural network training." International Conference on Machine Learning. PMLR, 2021.


**Summary Of The Paper:**

In this paper, the authors propose a new normalization method called MotifNorm for GNNs, which explicitly considers the intra-connection information and graph instance-specific statistics. The authors provide theoretical analyses to show that GNNs with MotifNorm are more expressive than the 1-WL test in distinguishing $k$-regular graphs. Experiments on several datasets are conducted to demonstrate the effectiveness of the proposed MotifNorm.

**Summary Of The Review:**

The current quality of the work (novelty, motivation, theoretical analysis, empirical results) is not ready for publishment in the venue, and I recommend rejection.

---

> ### Author Response · Authors · 2022-11-19
> **Response to reviewer iJ89 (Part-3)**
>
>
>
>
>
>
> **Q6. Runtime/Memory Cost evaluation should be provided for completeness. In equation 3, the structural weight need to count the number of edges and nodes in the node-induced subgraph, which increase additional computational costs for standard normalization methods. It is better to provide runtime and memory cost evaluation for comparison. Besides, the authors report the number of epochs that each model take to achieve the corresponding performance in Table 2. We can see that GNNs with MotifNorm converges slower than GNNs with BatchNorm in some cases. Thus, it is also reasonable to provide the total time that GNNs with different normalization methods take to achieve the reported scores.**
>
> **R6**.  The motif-induced weights **are preprocess and saved in the dataset**,  just like adjacency matrix and node features for loading. Thus, the increase additional computational cost mainly comes from RC and RE, where the time complex  is O(nd). To address your concern, we provide the runtime, parameter and memory comparison by using GCN (layer = 4) with BatchNorm and MotfiNorm on ogbg-molhiv dataset.
>
> |           | runtime     | parameters | memory |
> | --------- | ----------- | ---------- | ------ |
> | BatchNorm | 15.2s/epoch | 291.6K     | 2305M  |
> | MotifNorm | 22.6s/epoch | 293.2K     | 2347M  |
>
> ***On the page 9 of the revision, we supplied the above table.***
>
> Furthermore, the best epoch of MotifNorm is larger than BatchNorm does not mean that MotifNorm converges slower than GNNs with BatchNorm, and we provide the curve of the total time in **Figure 8** (**Page 18 of the revision**) for explanation.  Also, Figure 5 (b) can explain this phenomenon, where BatchNorm achieves the best performance at epoch 150.  In the meanhwile,  MotifNorm is better than BatchNorm and can still converge.
>
>
>
> **Summary of the response.**
>
> Thanks for your time and comments again.
>
> Accordingly to your comments,  we have carefully addressed your concerns from the perspective of motivation, novelty,  theoretical analysis and empirical results. At the same time, we have carefully polished the paper.
>
> We would be grateful if the reviewer could reconsider the score based on our clarifications and revisions, or let us know if further clarifications are required.

---

> ### Author Response · Authors · 2022-11-19
> **Response to reviewer iJ89 (Part-2)**
>
>
> **Q5. The results of baselines on several datasets are relatively lower than results in previous works. For example, for ZINC dataset, GCN with BatchNorm achieve 0.416 and 0.278 MAE respectively [3]. For ogbg-molhiv dataset, GCN with GraphNorm achieves 78.30 ROC-AUC [4]. For ogbn-proteins dataset, GCN with BatchNorm achieves 0.7251 ROC-AUC [3]. For ogbn-collab dataset, GraphSAGE with BatchNorm achieves 54.63% Hits@50 [3]. These results from previous works outperform the best results reported in this paper. Thus, the experimental results are doubtful and need to be clarified. Furthermore, many baselines are out-of-date and more recent methods such as graph transformers should be considered as baselines [If the proposed MotifNorm cannot bring SoTA results, why should people use it?].**
>
> **R5**. The performance of baselines are reported **in our paper without any tricks in GNNs' architecture and optimization**, and along with a unified parameter setting~(Table 9) on various datasets, **because such a comparison setting is straightforward for a general framework.**
>
> There are many tricks for GNNs, inculding Skip Connection (e.g., residual connection, jumping connection, init connection, etc.), Random Dropping (dropEdge, dropNode, etc.), adding additional information(e.g., virtual node, edge features, validation edges for inference, etc.), learning rate warmup, weight decay,  etc.  Secondly, there are many hyperparameters need to be tuned in the GNNs, such as batch size, learning rate, hidden dimesion, input feature and hidden feature dropout rate, etc.
>
> To address your concerns, we provide the corresponind results with some tricks in the following tables.
>
> (1). For ogbg-molhiv: **convolution with edge features**, input dropout as 0.0,  hidden dim 300, **weight decay in {5e-5, 1e-5}, residual connection**.
>
> (2). For ZINC dataset:  **input dropout as 0.0, hidden dropout as 0.0**, hidden dim 145, **residual connection**.
>
> (3). For ogbn-proteins:  **input dropout as 0.0, hidden dropout as 0.0.** hidden dim 256, num_layers = 2
>
> (4). For ogbl-collab: **Init connection**.
>
> |               | ogbg-molhiv       | ZINC              |               | ogbn-proteins     | ogbl-collab       |
> | ------------- | ----------------- | ----------------- | ------------- | ----------------- | ----------------- |
> | GCN+NoNorm    | 77.21 ± 0.430     | 0.473 ± 0.006     | GCN+PairNorm  | 69.84 ± 0.533     | 47.75 ± 0.190     |
> | GCN+UnityNorm | 77.56 ± 1.060     | 0.458 ± 0.009     | GCN+NodeNorm  | 72.53 ± 1.514     | 48.28 ± 1.100     |
> | GCN+ExpreNorm | 77.99 ± 0.545     | 0.436 ± 0.008     | GCN+MeanNorm  | 71.09 ± 1.236     | 47.27 ± 0.849     |
> | GCN+GraphNorm | 78.10 ± 1.115     | 0.396 ± 0.008     | GCN+GroupNorm | 73.17 ± 0.503     | 45.25 ± 1.206     |
> | GCN+BatchNorm | 78.07 ± 0.782     | 0.398 ± 0.003     | GCN+BatchNorm | 72.39 ± 0.611     | 49.44 ± 0.750     |
> | GCN+MotifNorm | **78.76 ± 0.371** | **0.358 ± 0.009** | GCN+MotifNorm | **73.55 ± 1.271** | **52.15 ± 0.647** |
>
> For your comment 'For ogbn-collab dataset, GraphSAGE with BatchNorm achieves 54.63% Hits@50 [3] (i.e., [2] in the following reference).'   we check the work [2] and do not find this result, and find you copy this result from ogb leaderboards[3].  It is worth mentioning that we do not use the val as input, thus, the results you provide may be 0.4714 (GCN+BatchNorm ) and 0.4810 (GraphSage+BatchNorm ).
>
> Furthermore, for the ZINC using GCN 16-layer: **Input dropout=0.0, hidden dropout=0.0, readout mlp(classifier) dropout=0.0, remove self-loop**, hidden dim 172, **residual connection.**
>
> |                  | ZINC                       |
> | ---------------- | -------------------------- |
> | GCN-16+BatchNorm | 0.27807973 ± 0.0050928     |
> | GCN-16+MotifNorm | **0.22857356 ± 0.0031086** |
>
> **At last. The baselines are different GNNs' normalization methods, which we have compared in the manuscript. Also graph transfomer, based on ViT paradigm, does not belong to the GNN paradigm, which is inappropriate to be mentioned in this work.**
>
> ***On page 8 of the revision, we updated the comparisons with empirical tricks in Table 5 and Table 6 to further improve the quality of the manuscript.***
>
>
> **Ref.**
>
> [1] Gao, Shang-Hua, et al. "Representative batch normalization with feature calibration." Proceedings of the IEEE/CVF Conference on Computer Vision and Pattern Recognition. 2021.
>
> [2] Dwivedi, Vijay Prakash, et al. "Benchmarking graph neural networks." arXiv preprint arXiv:2003.00982 (2020).
>
> [3] https://ogb.stanford.edu/docs/leader_linkprop/

---

> ### Author Response · Authors · 2022-11-19
> **Reoponse to  reviewer iJ89 (Part-1)**
>
> We thank the reviewer for the helpful suggestion.
>
> We are glad that the reviewer found the superiority of MotifNorm in experimental study.
>
> We will give the responses in order.
>
> **Q1. The motivation is not strong. The authors state that the designed weight factor needs at least to distinguish two subgraphs shown in Figure 2 (in page 3). However, it is not general to just use one pair of graphs as measurement. Such motivation is not convincing to support the following network design.**
>
> **R1**. Thanks for this comment.
>
> Two subgraphs shown in Figure 2 (in page 3) is just an example for explanation. In general, they are the **epitome of the subtree-isomorphic phenomenon**, where two subgraphs Sv1 and Sv2 are induced by root node v1 and v2 with the **same degree k** but the **connection information among neighborhoods is different.**  The subtree-isomorphic phenomenon prevalent in the real world decreases the GNNs’ expressivity in both the graph-level and node-level prediction. Thus, this is a general problem and we need to consider the ignored the node-induced subgraphs to compensate for the structural information.
>
> ***On page 1 of the revision, we have updated the above context in details.***
>
>
>
> **Q2. The novelty of the proposed MotifNorm is limited. The core step Representation Calibration (RC) of the proposed MotifNorm is highly related to the Representative Batch Normalization (RBN) [1], and the proof of Theorem 2 mainly follows [1]. The authors should carefully discuss the relation between RBN and MotifNorm.**
>
> **R2**. Sorry for the confusion.  And, we want to state that **the novelty of the Representation Calibration (RC) is to embed the structural information into normalization to compensate for the ignored characteristics by GNNs**, rather than balancing the distribution differences like Representative Batch Normalization (RBN). Thus, In order to prevent interference with judgment, we split this stability analysis in the **Proposition 1 of the revision (Page 5),** where we discuss the relation between RBN and MotifNorm. We hope reviewer iJ89 can reconsider the novelty of our paper, i.e., embed the ignored characterises into the normalization to compensate for the structural information for GNNs.
>
> ***On page 3-4 of the revision, we have updated the above context in details.***
>
>
>
> **Q3. Definitions and theorems presented in this paper are not rigorously written. In Definition 1, there do not exist definitions of the symbol u and u′. In Theorem 1, readers would be confused about which graphs the subgraphs Svi and Svj belong to (Please refer to [2] for formal statements). Moreover, both Theorem 2 and Theorem 3 are not quantitative analyses, which deliver little insightful information on the proposed MotifNorm.**
>
> **R3**. Many thanks for the insightful suggestion. We have rewritten the theorem with quantitative analyses in the revision.
>
> ***On the page 4 of the revision, we have provided the theorem with quantitative analyses.***
>
>
>
> **Q4. There exist mistakes in proofs for these theorems. For example, in Appendix A.2, the authors remove the MRC term to analyze the effect of representation calibration, however, it is non-trivial to conclude the final statements with the RC term. In Appendix A.3, the analysis on the over-smoothing issues is based on the equation 14, but this form largely differs from detailed implementations of the proposed MotifNorm. Thus, the correctness of theoretical analyses in this paper is doubtful.**
>
> **R4**. Thanks for the comment.  We added M_RC term to analyze the effect of representation calibration and the proofs for this theorem is correct, because that the values in M_RC are always positive numbers, which has no effect on the subsequent analysis. And we reprovided the analysis on the over-smoothing issue when keep consistent with experimental details in the Theorem 1 of the revision.
>
> ***On page 4-5 of the revision,  we updated the above context as the theorem 1 and proposition 1.***
>
> **Ref.**
>
> [1] Gao, Shang-Hua, et al. "Representative batch normalization with feature calibration." Proceedings of the IEEE/CVF Conference on Computer Vision and Pattern Recognition. 2021.
>
> [2] Dwivedi, Vijay Prakash, et al. "Benchmarking graph neural networks." arXiv preprint arXiv:2003.00982 (2020).
>
> [3] https://ogb.stanford.edu/docs/leader_linkprop/

---

> ### Author Response · Authors · 2022-11-26
> **Looking forward to your further feedback**
>
> Dear Reviewer iJ89,
>
> We sincerely thank you for the valuable suggestions that have helped us improve the quality of the paper significantly. Our work attempts to design a powerful normalization module to compensate for the sub-structural characteristics ignored by the message-passing GNNs.
>
> We hope our responses from the perspective of motivation, novelty, theoretical analysis, and empirical results could address your concerns.  And we would really appreciate the opportunity to discuss this further if our responses have not already addressed your concerns.
>
> Yours sincerely,
>
> Authors

---

### Official Review · Reviewer_N94g · 2022-10-24

**Confidence:** 3
**Correctness:** 3
**Technical Novelty And Significance:** 2
**Empirical Novelty And Significance:** 2
**Recommendation:** 3

**Clarity, Quality, Novelty And Reproducibility:**

Many parts of the paper are unclear, e.g., many of the formal statements are of a too handwavey nature. The proposed normalization layer seems to be derived in an ad-hoc manner.

**Strength And Weaknesses:**

**Strengths**
- Well done experimental study
- Simple approach that seems to give some boosts to standard datasets and standard GNN architectures


**Weaknesses**
- The design of the normalization layer is not well motivated, the design seems adhoc
- Theorem statements/proofs are of a very handwavey nature, not clear  and not rigorous enough

**Suggestions, issues, questions, and comments**
- The formal statement of Theorem 1 is unclear. What is meant by the sentence after "while any"?
- Equation 2: This is a very specific GNN layer. Note that the layer does not use a non-linearity.
- Over-smoothing was only shown for very specific GNNs, it is not well understood if this holds for all possible GNN architectures. For example, 1-WL-equivalent GNNs, at least in theory, will not suffer from over-smoothing as the vertex partition gets finer as one increases the number of layers/iterations
- Section 3: It should be made more clear why it is essential to include the structural information, encoding the neighborhood subgraph into the normalization. Why not just add it as an extra feature to the node features?
- Section 3.1. before Theorem 2: The motivation/explanation of MotifNorm should be improved. This seems to be the main contribution but it is hard to follow although it seems to be quite simple
- Below Equation (5): It is not clear what the term "segment averaging" refers to
- Equation (4): Do all normalization layers fit in the provided form? Can you provide a reference for this?

- Theorem 2: The statement is not formal enough. For example, it is unclear what "stabling and accelerating GNNs’ training" means.
- Theorem 3: The proof sketch is not helpful, e.g., make at least clear why GNNs+MotifNorm can distinguish the two graphs. Moreover, the statement "helps alleviate the over-smoothing issue when GNNs’ layers become deeper" is not formal enough for a theorem statement.
- Section 4.1: What is the reason for performing a "hierarchical dataset splitting strategy based on the structural statistics of graphs". Is this necessary to get good results?
- Table 3: The reported numbers for ZINC seem very high, e.g., compare to https://arxiv.org/abs/2003.00982. What is the reason for this?

**Minor points**
- Page 2: Your edges are directed as they are from $V_G \times V_G$
- Section 2.1: "Graph isomorphism problem" might be the better term ("isomorphism issue")
- Equation 1: Define the initial labeling, it might also be helpful to formally define the considered GNNs in the main paper
- Definition 2: "number of samples in a countable set" -> "cardinality"

**Summary Of The Paper:**

The paper deals with the design of normalization layers for GNNs that boost their expressivity and prevent over-smoothing. To that, the authors propose a normalization layer that takes structural information, namely graph structure induced by the direct neighbors of a given vertex, into account. The expressivity and its capabilities with regard to the prevention of over-smoothing are investigated theoretically. Further, the new layer is evaluated empirically, on a large set of problems and datasets, showing promising performance.




**Summary Of The Review:**

The paper proposes a simple normalization layer that boosts standard GNNs' empirical performance on well-known benchmark datasets. However, the design of the layer seems ad-hoc, and the theoretical explanation lacks rigor. In its current form, the paper is not ready for a top-tier conference.

---

> ### Author Response · Authors · 2022-11-19
> **Response to reviewer N94g (Part-3)**
>
>
>
>
> **Q12. Section 4.1: What is the reason for performing a "hierarchical dataset splitting strategy based on the structural statistics of graphs". Is this necessary to get good results?**
>
> **R12**. The reason is that this strategy can make the training, valid and test sets follow the same distribution as possible, which is essential to report the performance.  If the valid and test set obey different distribution, we select the best epoch on the valid set, where may be not the best epoch on the test set.
>
> **Response to `Is this necessary to get good results?' :**  It is designed to obtain more fair and reasonable results.
>
> ***On page 5 of the revision,  we highlighted this reason in the Section 4.1.***
>
>
>
> **Q13. Table 3: The reported numbers for ZINC seem very high, e.g., compare to https://arxiv.org/abs/2003.00982. What is the reason for this?**
>
> **R13**. The reason is that the results for ZINC regression in https://arxiv.org/abs/2003.00982 are reported **by using residual connection and without feature dropout.**  We provided the comparison results in the first version without any tricks in GNNs.
>
> To address your concern and further improve the experimental results, **we set input dropout as 0.0, hidden dropout as 0.0, hidden dim as 145, and add residual connection to obtain**
>
> |               | ZINC              |
> | ------------- | ----------------- |
> | GCN+NoNorm    | 0.473 ± 0.006     |
> | GCN+UnityNorm | 0.458 ± 0.009     |
> | GCN+ExpreNorm | 0.436 ± 0.008     |
> | GCN+GraphNorm | 0.396 ± 0.008     |
> | GCN+BatchNorm | 0.398 ± 0.003     |
> | GCN+MotifNorm | **0.358 ± 0.009** |
>
> ***On page 8 of the revision, we supplied the results with empirical tricks in Table 5 and Table 6 to further comparisons.***
>
>
>
> **Q14. Minor points: (1) Page 2: Your edges are directed as they are from VG×VG. (2)Section 2.1: "Graph isomorphism problem" might be the better term ("isomorphism issue"). (3) Equation 1: Define the initial labeling, it might also be helpful to formally define the considered GNNs in the main paper. (4) Definition 2: "number of samples in a countable set" -> "cardinality"**
>
> **R14**. Thanks for these kind suggestions, we have corrected the above problem **in the revision (mainly on page 2, 3).**
>
>
>
> **Summary of the response.**
>
> Thanks for your time and comments again.
>
> Accordingly to your comments, we have carefully revised paper and give the above responses.
>
> We would be grateful if the reviewer could reconsider the score based on our clarifications and revisions, or let us know if further clarifications are required.
>
>
> Ref.
>
> [1] Uncovering the Structural Fairness in Graph Contrastive Learning. NeurIPS, 2022.
>
> [2] How Powerful are Spectral Graph Neural Networks. ICML, 2022.
>
> [3] Convolutional Neural Networks on Graphs with Chebyshev Approximation, Revisited. NeurIPS, 2022.
>
> [4] Pairnorm: Tackling oversmoothing in gnns. ICLR, 2020.
>
> [5] Batch normalization: Accelerating deep network training by reducing internal covariate shift. ICML, 2015.

---

> ### Author Response · Authors · 2022-11-19
> **Response to reviewer N94g (Part-2)**
>
>
>
> **Q6. Section 3: It should be made more clear why it is essential to include the structural information, encoding the neighborhood subgraph into the normalization. Why not just add it as an extra feature to the node features?**
>
> **R6**. **The reason of embedding subgraph information:** The ignore of node-induced information limits the expressive capability of GNNs to **address subtree-isomorphic phenomenon** prevalent in the real world, which decreases the GNNs’ expressivity in graph-level and node-level prediction, i.e., (1) **Graph-level:** Straightforward neighborhood aggregations, ignoring the characterises of the node-induced subgraphs, result in a complete indistinguishability in the subtree-isomorphic case, which thus leads the GNNs’ expressivity to be bottlenecked by the WL test. (2) **Node-level:** Under the background of oversmoothing, the smoothing problem among the root representations of subtree-isomorphic substructures will become worser when aggregating the similar representations from their neighborhoods without structural characterises be considered.
>
> **Response to `Why not just add it as an extra feature to the node features?'** Firstly, structural weighs are the scalar values and do not exist in the same distribution space as node features, it is not suitable for adding as the extra feature.  Secondly,  we strive to develop a **general framework** to improve the GNNs' expressivitiness. Thirdly,  deep models usually follow convolution, normalization and activation architecture,  where the normalization module generally follows GNNs convolution operation.
>
> ***On Page 1 of the revision,  the context mentioned above has been updated in the Section 1.***
>
>
>
> **Q7. Section 3.1. before Theorem 2: The motivation/explanation of MotifNorm should be improved. This seems to be the main contribution but it is hard to follow although it seems to be quite simple**
>
> **R7**. Thanks for your kind comment. We have added the insight explanation of MotifNorm **in the revision (Page 3-4).**
>
>
>
> **Q8. Below Equation (5): It is not clear what the term "segment averaging" refers to**
>
> **R8**. H_SA is the segment averaging of H, obtained by sharing the average node features of each individual graph with its nodes, where each
> individual graph is called a segment in the DGL implementation.
>
> ***On page 4 of the revision, the context mentioned above has been updated.***
>
>
>
> **Q9. Equation (4): Do all normalization layers fit in the provided form? Can you provide a reference for this?**
>
> **R9**. Not all norm techniques fit this formula, However, the Equation (4) is the formula of the first norm method, i.e., BatchNorm [5].  And, the most of norms methods follows this  formula.
>
>
>
> **Q10. Theorem 2: The statement is not formal enough. For example, it is unclear what "stabling and accelerating GNNs’ training" means.**
>
> **R10**. Thanks for your kindly comment.  Stable means that MotifNorm with RC can balance the distribution differences between the graph-instance and batch. Accelerate means that MotifNorm can help GNNs converge quickly.
>
>  ***On page 5 of the revision, the context mentioned above has been updated in the Proposition 1.***
>
>
>
> **Q11. Theorem 3: The proof sketch is not helpful, e.g., make at least clear why GNNs+MotifNorm can distinguish the two graphs. Moreover, the statement "helps alleviate the over-smoothing issue when GNNs’ layers become deeper" is not formal enough for a theorem statement.**
>
> **R11**. Thanks for your kind comment.  We have rewritten the Expressivity Analysis of the MotifNorm in the revision.
>
> ***On page 4 of the revision, we updated the Expressivity Analysis of the MotifNorm for graph-level and node-level tasks. Furthermore, the theorem analysis about the over-smoothing issue is updated as Theorem 1.***
>
> Ref.
>
> [1] Uncovering the Structural Fairness in Graph Contrastive Learning. NeurIPS, 2022.
>
> [2] How Powerful are Spectral Graph Neural Networks. ICML, 2022.
>
> [3] Convolutional Neural Networks on Graphs with Chebyshev Approximation, Revisited. NeurIPS, 2022.
>
> [4] Pairnorm: Tackling oversmoothing in gnns. ICLR, 2020.
>
> [5] Batch normalization: Accelerating deep network training by reducing internal covariate shift. ICML, 2015.

---

> ### Author Response · Authors · 2022-11-19
> **Response to reviewer N94g (Part-1)**
>
> We thank the reviewer for the comments.
>
> We are glad that the reviewer found the superiority of MotifNorm in experimental study.
>
> Please see below comments for your concerns.
>
> **Q1. The design of the normalization layer is not well motivated, the design seems adhoc**.
>
> **R1**.  We give the details of motivation as follows:
>
> **Firstly**,  The motivation is that the existing GNNs usually follow the neighborhood aggregation scheme, ignoring the structural characteristics in the node-induced subgraphs, which limits their expressiveness to address **subtree-isomorphic phenomenon** prevalent in the downstream tasks of both the graph- and node-level predictions.  Thus, we strive to develop a general framework, **compensating for the ignored characteristics among the node-induced subgraphs**, to improve the graph expressivity over the prevalent message-passing GNNs.
>
> **Secondly**, Driven by the fact that deep models usually follow the convolution, normalization and activation architecture, where the normalization module generally follows GNNs convolution operations, we thus accordingly focus on embedding the ignored structural information in normalization module to generally improve the expressivity for GNNs. **Furthermore,  the existing normalization approaches are usually task-specific in GNNs’ architectures, which means that they are not always significant in general contribution to various downstream tasks, and ignore  the characteristics of node-induced substructures.**
>
> **Thirdly**, We design the Representation Calibration and Representation Enhancement to embed the ignored structural information for GNNs.   **Representation Calibration** calibrates the inputs to balance the distribution differences along with embedding structural information.  **Representation Enhancement**  ensures that motif-induced weights is embedded to maintain the structural information in the final representation.
>
> ***On Page 1-2 of the revision,  the context mentioned above has been updated in the Section Abstract and Introduction.***
>
>
>
> **Q2. Theorem statements/proofs are of a very handwavey nature, not clear and not rigorous enough**
>
> **R2**. Thanks for this comment,  we have carefully polished the theorem/proofs, and re-provide the theorem 1 and proposition 1 ***on page 4-5 of the revised version.***
>
>
>
> **Q3. The formal statement of Theorem 1 is unclear. What is meant by the sentence after "while any"?**
>
> **R3**. Here means that *`For any two different subgraphs S_{v_i} , S_{v_j} are subtree-isomorphic, GNNs are as powerful the as 1-WL test in distinguishing non-isomorphic graphs'* .  To make the article more clear, we delete the original theorem 1 in the revised version.
>
>
>
> **Q4. Equation 2: This is a very specific GNN layer. Note that the layer does not use a non-linearity.**
>
> **R4**. The over-smoothing problem is mainly related to the GNN message-passing paradigm. Moreover, non-linearity layer compresses the value domain, which makes GNNs more prone to over-smoothing issue.  Furthermore, works [1-3] demonstrate that the message-passing GNNs work as a low-pass filter, which preserves the low-frequency information, i.e., the information that are similar. This provides the explanation of over-smoothing phenomenon from another perspective.
>
>
>
> **Q5. Over-smoothing was only shown for very specific GNNs, it is not well understood if this holds for all possible GNN architectures. For example, 1-WL-equivalent GNNs, at least in theory, will not suffer from over-smoothing as the vertex partition gets finer as one increases the number of layers/iterations**
>
> **R5**. Over-smoothing issue is common in the **message-passing GNNs** (i.e., Spatial GNNs) [1-4] and the above example is not appropriate.  Assume there are  two graphs (four nodes in each graph) with different structures,  1-WL equivalent GNNs can distinguish two **structures** by labeling them as **{red, green, yellow, black}** and **{red, yellow, blue,  pink}**. However, **in each graph,**  the values mapped from **{red, green, yellow, black}** or **{red, yellow, blue,  pink}** will become similar when increasing the number layers, even if their values are not consistent. That is to say, **over-smoothing exist in a graph, rather than two different graphs.**
>
>
> Ref.
>
> [1] Uncovering the Structural Fairness in Graph Contrastive Learning. NeurIPS, 2022.
>
> [2] How Powerful are Spectral Graph Neural Networks. ICML, 2022.
>
> [3] Convolutional Neural Networks on Graphs with Chebyshev Approximation, Revisited. NeurIPS, 2022.
>
> [4] Pairnorm: Tackling oversmoothing in gnns. ICLR, 2020.

---

> ### Author Response · Authors · 2022-11-26
> **Looking forward to your further feedback**
>
> Dear Reviewer N94g,
>
> We sincerely thank you for the valuable suggestions that have helped us improve the quality of the paper significantly.  Our work attempts to design a powerful normalization module to compensate for the sub-structural characteristics that ignored by the message-passing GNNs.
>
> We hope our responses to the theorem statements and MotifNorm's motivation could address your concerns.  And we would really appreciate the opportunity to discuss this further if our responses have not already addressed your concerns.
>
> Yours sincerely,
>
> Authors

---

### Author Response · Authors · 2022-11-19
**Summary for major changes**

Dear AC and reviewers,

We sincerely appreciate your valuable time and constructive comments. We have uploaded a revised version of our paper, where major changes are highlighted in blue. Below is the summary of major changes:

1. We revise Section introduction (page 1-2), which may cause confusion about the more general motivation of MotifNorm.
2. We add the section Related Work to introduce and discuss the recent normalization works for GNN (page 2-3).
3. We add the insight explanation with respect to Representation Calibration and Representation Enhancement operation (page 4).
4. We polish the theorem analysis and proposition (page 4-5).
5. We provide the results to show that MotifNorm can further improve the performance of GSN/GraphSNN (page 6).
6. We provide the comparison with empirical tricks to further improve the performance for various tasks (page 8).
7. We provide the Runtime/Memory Cost evaluation of MotifNorm (page 9).

We sincerely hope our responses and revisions address all reviewers’ concerns, and believe these updates may help us better deliver the benefits of the proposed MotifNorm to the ICLR community.

Thank you very much,

Authors.

---

### Author Response · Authors · 2022-12-04
**Summary for clearer discussion**

Dear AC and Reviewers,

Many thanks for your valuable time and constructive comments.

We provide the significant changes in the revision that address reviewers' main concerns for clearer discussion:

**Q1.** the generalization~(reviewer wzdD concerned)

**R1.** The commonly used message-passing GNNs are weaker than the 1-WL test, and there are no quantitative analyses to measure their expressiveness gap, which makes the expressiveness analysis between an arbitrary GNN+ MotifNorm and 1-WL test difficult. To address the reviewer's concern, we try to give the expressiveness analysis from the perspective of addressing a more general case, i.e., **subtree-isomorphic**, which is a more general phenomenon in real applications. MotfiNorm compensates for the structural information to distinguish the subtree-isomorphic case that 1-WL can not recognize, and thus generally improve the graph expressiveness. Specially, we give the two cases of GNN+ MotifNorm beyond the 1-WL test (1). GNN+ MotifNorm beyond 1-WL in the case of k-regular graphs. (2) GNN+ MotifNorm will go beyond the 1-WL test in an arbitrary graph classification task when the adopted GNNs are 1-WL-equivalent GNNs. e.g., GIN.



**Q2.** the reproducibility~(reviewer WzdD, iJ89 and N94g concerned)

**R2.** The performance of baselines is reported **in our paper without any tricks in GNNs' architecture and optimization**. To address reviewers' concerns, we provide the corresponding results with some GNNs' tricks in Table 5 and Table 6 of the revision, where the results can be obtained by using our submitted codes. For example, argument '--econv' for adding edge features, '--skip_type' set as 'Residual' or 'Initial' for the empirical skip connection, etc.



**Q3.** the motivation~(reviewer iJ89  and N94g concerned)

**R3.** The motivation is that the existing GNNs usually follow the neighborhood aggregation scheme, ignoring the structural characteristics in the node-induced subgraphs, which limits their expressiveness to address **the subtree-isomorphic phenomenon** prevalent in the downstream tasks of both the graph- and node-level predictions. Thus, we strive to develop a general framework, **compensating for the ignored characteristics among the node-induced subgraphs**, to improve the graph expressivity over the prevalent message-passing GNNs.

***On Page 1-2 of the revision, the context mentioned above has been updated in the Section Abstract and Introduction.***



**Q4.** the novelty~(reviewer iJ89 concerned)

**R4.**  The existing norms are usually task-specific in GNNs’ architectures, which means that they are not always significant in general contribution to various downstream tasks. Furthermore, the characteristics of node-induced substructures, which trouble GNNs’ performance in various downstream tasks, are ignored by these normalizations. Thus, **the novelty of our work is to embed the structural information into normalization to compensate for the ignored characteristics by GNNs, which  improves the graph expressivity over the prevalent message-passing GNNs for various graph downstream tasks, e.g., graph, node and link predictions.**

We look forward to an opportunity to discuss our carefully revised manuscript before the second discussion ends.

Yours sincerely,

Authors

---

### Decision · Program_Chairs · 2023-01-20

**Decision:**

Reject

**Justification For Why Not Higher Score:**

- There are many works attempting to 1) address oversmoothing and 2) include motif into GNNs. Authors need to find a better position of their work and motivate their design more clearly.
- Authors also need to show the theoretical results in a more mathematical and formal way.

**Justification For Why Not Lower Score:**

N/A

**Metareview: Summary, Strengths And Weaknesses:**

Graph neural networks (GNNs) are known to suffer from over-smoothing issue, this paper proposes to solve this issue by adding a plug-and-play layer, i.e., motif-induced normalization layer, into GNNs. Theoretical analysis has been done, which shows over-smoothing problem can be alleviated, and the proposed operation can stabilize model training. Extensive experiments have been done to demonstrate the effectiveness of the proposed method.

Strengths
- Extensive experiments have been done.
- Simple approach which seems can be combined with many existing GNNs.

Weaknesses
- Motivation of the design is not convincing. More discussion with existing works should be included.
- Theoretical statements are informal and not rigorous. The proofs need to be double-checked as well.